# Sequential mutations in exponentially growing populations

**Michael D. Nicholson**[1]*, **David Cheek**[2], **Tibor Antal**[3]

**1** Edinburgh Cancer Research, Institute of Genetics and Cancer, University of Edinburgh, Edinburgh, United Kingdom, **2** Center for Systems Biology, Department of Radiology, Massachusetts General Hospital Research Institute and Harvard Medical School, Boston, Massachusetts, United States of America, **3** School of Mathematics and Maxwell Institute for Mathematical Sciences, University of Edinburgh, Edinburgh, United Kingdom

\* mdnicholson5@gmail.com

## Abstract

Stochastic models of sequential mutation acquisition are widely used to quantify cancer and bacterial evolution. Across manifold scenarios, recurrent research questions are: how many cells are there with $n$ alterations, and how long will it take for these cells to appear. For exponentially growing populations, these questions have been tackled only in special cases so far. Here, within a multitype branching process framework, we consider a general mutational path where mutations may be advantageous, neutral or deleterious. In the biologically relevant limiting regimes of large times and small mutation rates, we derive probability distributions for the number, and arrival time, of cells with $n$ mutations. Surprisingly, the two quantities respectively follow Mittag-Leffler and logistic distributions regardless of $n$ or the mutations' selective effects. Our results provide a rapid method to assess how altering the fundamental division, death, and mutation rates impacts the arrival time, and number, of mutant cells. We highlight consequences for mutation rate inference in fluctuation assays.

**Data Availability Statement:** Code and data relating to this study can be found at www.github.com/MichaelDNicholson/accumulate_nmutations.

**Funding:** M.D.N is a cross-disciplinary post-doctoral fellow supported by funding from CRUK Brain Tumour Centre of Excellence Award (C157/

## Author summary

In settings such as bacterial infections and cancer, cellular populations grow exponentially. DNA mutations acquired during this growth can have profound effects, e.g. conferring drug resistance or faster tumour growth. In mathematical models of this fundamental process, considerable effort—spanning many decades—has been invested to understand the factors that control two key aspects of this process: how many cells exist with a set of mutations, and how long does it take for these cells to appear. In this paper, we consider these two aspects in a general mathematical framework. Surprisingly, for both quantities, we find universal probability distributions which are valid regardless of how many mutations we focus on, and what effect these mutations might have on the cells. The distributions are elegant and easy to work with, providing a computationally efficient alternative to intensive simulation-based approaches. We demonstrate the

A27589). The funders played no role in the study design, data collection and analysis, decision to publish, or preparation of the manuscript.

**Competing interests:** No competing interests to declare.

usefulness of our mathematical results by illustrating their consequences for bacterial experiments and cancer evolution.

## Introduction

To quantitatively characterise diseases, in settings such as cancer, and bacterial and viral infections, a concerted effort has been made to study evolutionary dynamics in exponentially expanding populations. Understanding the timescale of evolution is a key aspect of this research program which has proven useful in a diverse range of areas such as: measuring mutation rates [1], assessing the likelihood of therapy resistance developing [2–4], inferring the selective advantage of cancer driver events [5–7], and exploring the necessary steps in the metastatic process [8, 9]. The common theme within these works is that they use information about when a particular cell type arises within the population of interest. For a concrete example, whose roots lie in the celebrated work of Luria and Delbrück [1], if we imagine a growing colony of bacteria, we might wish to know how quickly a mutant bacterium will develop with a specific mutation that confers resistance to an antibiotic therapy.

The time until a cell type emerges, and expands to a detectable population size, depends on a variety of factors. Most obvious are the relevant mutation rates, however selection also plays an important role. For instance, if we start an experiment with an unmutated cell and wait for a cell with 2 mutations, a low division rate of cells with one mutation slows down this process. In the scenario of the sequential acquisition of driver alterations in cancer, with each mutation providing a selective advantage, Durrett and Moseley characterised the time to acquire $n$ driver mutations [10]. We recently examined the setting of drug resistance conferring mutations, which often have a deleterious effect, so that the original cell type grew the fastest [11]. However, in general, the effects of mutation and selection on evolutionary timescales within exponentially growing populations remain unclear.

In this study we build upon the mathematical machinery developed in Refs. [10, 11] to investigate this question. We focus on the biologically relevant settings of large times and small mutation rates. Broad-ranging features of the cell number, and arrival time, of type $n$ cells are highlighted—including universal simple distributions—and explicit expressions make the impact of mutation and selection clear.

## Model

### Model

We consider a population of cells, where each cell can be associated with a given 'type' (for example 'type 3' might be cells with 3 particular mutations). Cells of type $n$ divide, die, and mutate to a cell of type $n + 1$, at rates $\alpha_n$, $\beta_n$ and $v_n$, with all cells behaving independently of each other. With ($n$) representing a type $n$ cell and $\varnothing$ symbolising a dead cell, our cell level dynamics can be represented as (see also Fig 1A):

$$(n) \rightarrow \begin{cases} (n), (n) & \text{at rate } \alpha_n \\ \varnothing & \text{at rate } \beta_n \\ (n), (n+1) & \text{at rate } v_n. \end{cases} \tag{1}$$

In other words after a random, exponentially distributed waiting time with parameter $\alpha_n + \beta_n + v_n$, a type $n$ cell is replaced by one of the listed three options with probability proportional to its corresponding rate. The process starts with a single cell of type 1 at time $t = 0$, and we

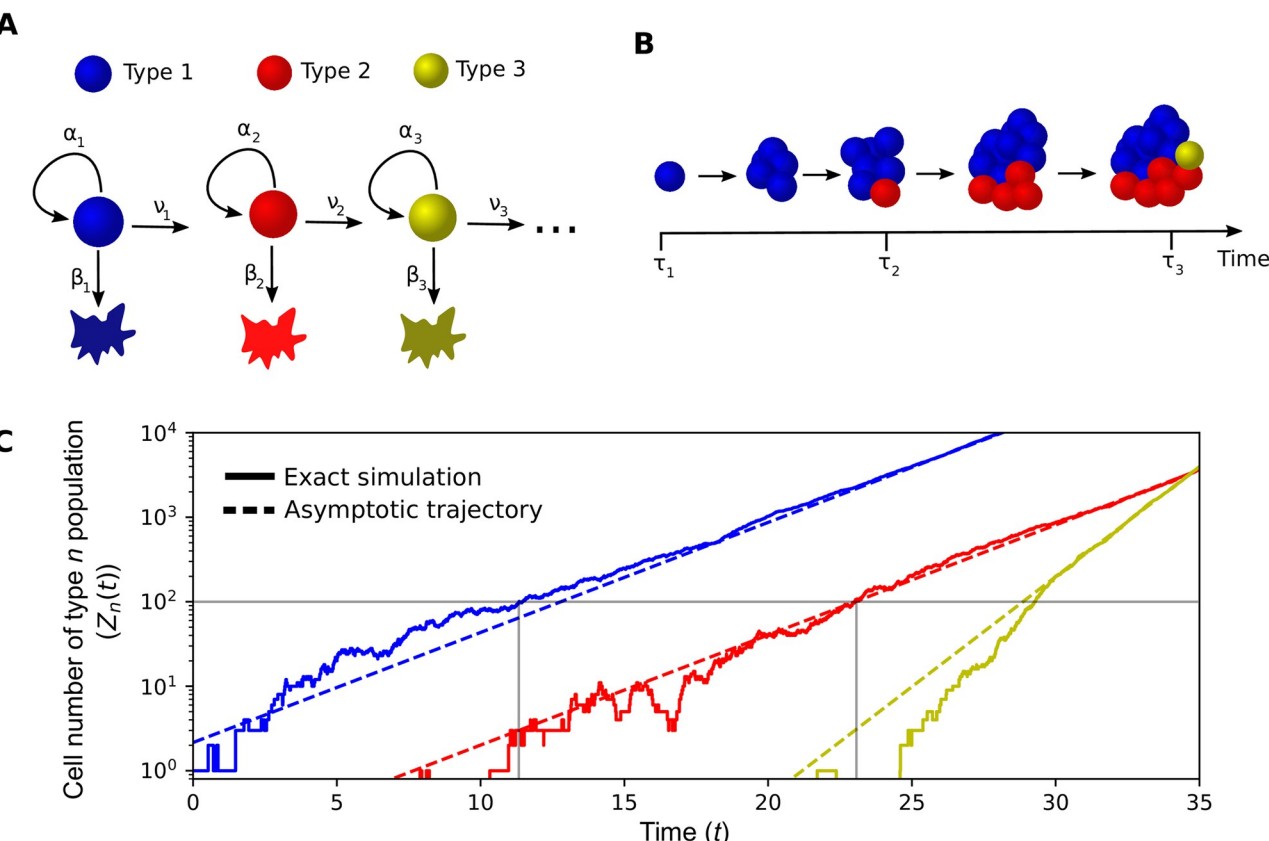

**Fig 1. Model schematic.** A: We consider a multitype branching process in which cells can divide, die, or mutate to a new type. B: We study the waiting time until a cell of the $n$th type exists, $\tau_n$, starting with a single cell of type 1. C: Stochastic simulation of the number of cells over time, with dashed lines indicating the large-time trajectories given by Eq (1). Grey horizontal line occurs at the inverse of the mutation rate, while the grey vertical lines indicate the time at which the type $n$ population size reaches the inverse of the mutation rate, which gives the arrival time of the type $n + 1$ cells to leading order. Parameters: $\alpha_1 = \alpha_3 = 1.1$, $\alpha_2 = 1$, $\beta_1 = 0.8$, $\beta_2 = 0.9$, $\beta_3 = 0.5$, $\nu_1 = \nu_2 = 0.01$. Thus, the net growth rates are $\lambda_1 = 0.3$, $\lambda_2 = 0.1$, $\lambda_3 = 0.6$ and the running-max fitness follows $\delta_1 = \delta_2 = \lambda_1$, $\delta_3 = \lambda_3$.

assume that the type 1 population is supercritical ($\alpha_1 > \beta_1$) and that it survives forever (does not undergo stochastic extinction).

We focus on two quantities; the number of cells of type $n$ at time $t$—denoted $Z_n(t)$, and the arrival time of the first type $n$ cell—termed $\tau_n$ (see Fig 1B and 1C). To describe the growth of the cellular populations, let the net growth rate of the type $n$ cells be $\lambda_n = \alpha_n - \beta_n$. We denote the 'running-max' fitness, which is the largest growth rate of the cell types among $1, \ldots, n$, as $\delta_n$, that is $\delta_n = \max_{i=1,\ldots,n} \lambda_i$. Further, we introduce $r_n$ as the number of times the running-max has been attained over the cell types up to $n$, that is $r_n = \#\{i = 1, \ldots, n : \lambda_i = \delta_n\}$. A summary of the key notation used in this article is provided in Table 1.

## Motivation

Our model considers a linear evolutionary path of cells sequentially mutating from type 1 to 2 to 3, and so on (see Figs 1A and 2). We briefly highlight scenarios for which our model is relevant, drawing on examples from cancer evolution (although similar statements can be made for other exponentially growing populations).

Cancer cells accumulate mutations with a variety of phenotypic effects during the cancer's expansion. Oncogenic driver mutations are thought to increase the population's net growth

**Table 1. Key notation used throughout this article.**

| Notation | Description |
|---|---|
| $\alpha_n, \beta_n$ | Division and death rate of type $n$ cells |
| $\lambda_n$ | Net growth rate of type $n$ cells, i.e. $\alpha_n - \beta_n$ |
| $\nu_n$ | Mutation rate of type $n$ cells |
| $\delta_n$ | Running-max fitness, i.e. $\max_{i=1,\ldots,n}\{\lambda_i\}$ |
| $r_n$ | Number of times the running-max fitness has been attained over types $1,\ldots,n$, i.e. $\#\{i = 1,\ldots,n : \lambda_i = \delta_n\}$ |
| $Z_n(t)$ | Cell number of type $n$ at time $t$ |
| $\tau_n$ | Arrival time of type $n$ cells |
| $t_{1/2}^{(n)}$ | Median arrival time of type $n$ cells |
| $V_n$ | 'Random amplitude' of approximate cell number of type $n$ (see Eq (1)) |
| $\omega_n$ | Scale parameter of 'random amplitude' (see Eq (2)) |

rate, either by increasing the proliferation rate or decreasing the death rate. A linear path is relevant when considering cancers that follow a specified evolutionary trajectory. For example, the canonical mutational path [12, 13] in colorectal cancer is loss of *APC* (type 1 cells), followed by a *KRAS* mutation (type 2 cells have mutations in both genes), then loss of *TP53* (type 3 cells with mutations in all 3 genes); see Fig 2B.

When the cancer evolutionary trajectory is not specified, but it is assumed that driver mutations arise at a constant rate such that each new mutation confers a constant $1 + s_d$ fold

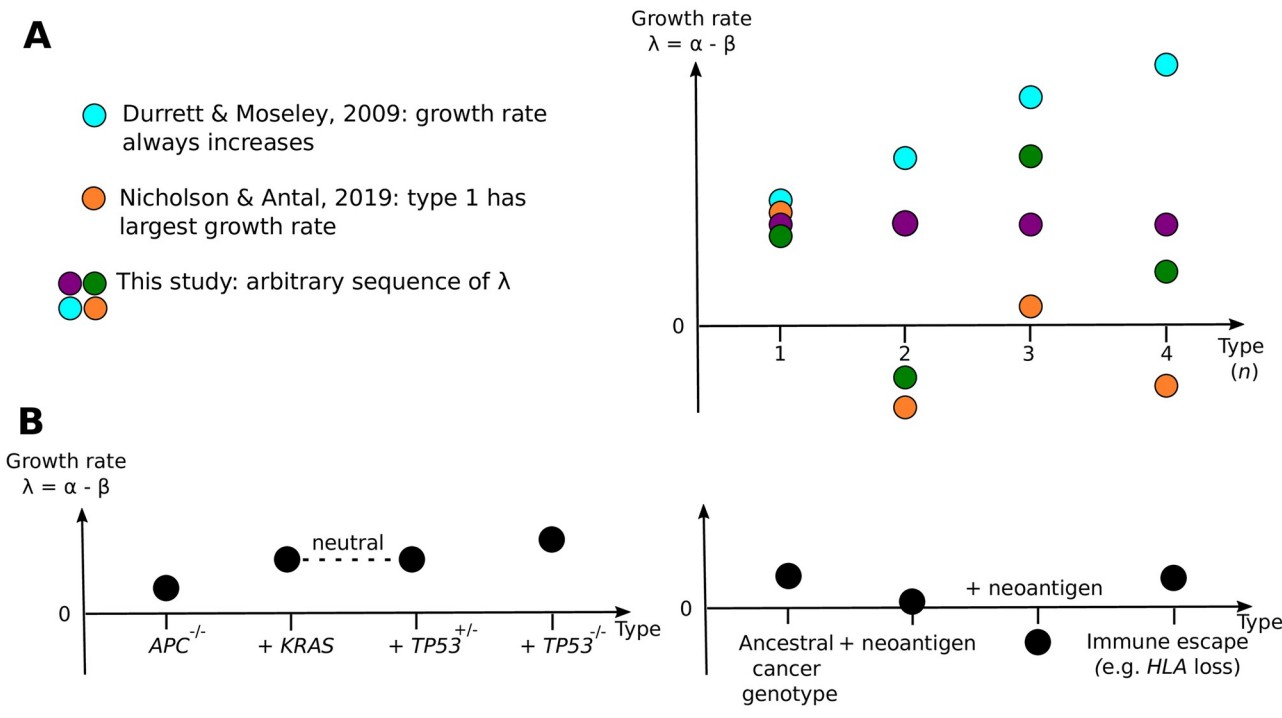

**Fig 2. Comparison with prior work and motivating examples.** A. Previous work has considered special cases of growth rate sequences, here we consider general sequences as long as $\lambda_1 > 0$. B. Two biological scenarios in which the growth rate sequences covered in this paper are relevant: the acquisition of driver mutations in the canonical carcinogenesis pathway of colorectal cancer, and the accumulation of neoantigens by cancer cells which results in increased cell death due to immune system surveillance.

increase in the proliferation rate, then this model also falls within our framework. Bozic *et al.* [5] applied this model to cancer genetic data, thereby inferring the selective effect $s_d$ of driver mutations. Conversely to oncogenic drivers, neoantigen-creating mutations that stimulate the immune system to attack cancer cells have been modelled as increasing the death rate of the mutated cells by a factor of $1 + s_n$ [14] (Fig 2B). Lakatos *et al.* [14] used this model to examine conditions such that a population of neoantigen-presenting cancer cells would be sufficiently large to be observed in sequencing data in order to explore the limits of detecting immune-mediated negative selection. Exploring how the distribution of the cell number with $k$ neoantigens varies as function of $s_n$ and the neoantigen-mutation rate can be rapidly assessed with the results below.

For a more general model that describes a population with the potential to traverse multiple evolutionary paths, genotype space can be represented as a directed graph. When the original cell type has the largest net growth rate, we recently derived simple formulas for the arrival time and cell number through the directed graph of genotypes [11]. The results presented below, where the cell type with the largest net growth rate is unconstrained, hold only for a linear path through a genotype space. While in this work we cannot compare arbitrary sets of paths to a target evolutionary genotype, one may focus on each evolutionary path to the target type separately as a single linear path and then compare the median time to traverse each evolutionary path using the results presented below. For example, two sets of driver mutations might be considered: mini-drivers which have a high mutation rate, but low selective advantage, and major-drivers which have a low mutation rate but large selective advantage [15]. We would then compare the median times of the evolutionary paths 'Driver 1 → Mini-driver → Driver 3' and 'Driver 1 → Major-driver → Driver 3' to determine which path is most likely to produce the first cell with three driver mutations.

The cancer evolution examples discussed above all assume that the type 1 cell has a driver mutation. In other settings, it may be more natural to consider the type 1 cells as wild type, for example when considering the emergence of drug resistance. We emphasise that in this paper the type one cells are always supercritical, that is they grow exponentially on average.

## Results

Our results are broken into three sections. We first give an overview of our main mathematical results, stratified by whether they relate to the number of type $n$ cells or to their arrival time. We then highlight the main properties of the results as well as providing intuitive arguments for why these properties emerge. Finally, we compare our results to previously known special cases.

### Results overview

**Population sizes.** Understanding the distribution of the number of cells of type $n$ at a fixed time $t$ (e.g. the probability that 5 cells exist of type 2 at time 2) can be complex [16], however a surprising level of simplicity emerges at large times with small mutation rates. The number of cells of type $n$ can be decomposed into the product of a time-independent random variable and a simple time-dependent deterministic function controlled by the running-max fitness $\delta_n$, and the number of times it has been attained $r_n$ up to type $n$:

$$Z_n(t) \approx V_n t^{r_n - 1} e^{\delta_n t}. \tag{1}$$

The random variable $V_n$ has a Mittag-Leffler distribution with tail parameter $\lambda_1/\delta_n$, and scale parameter $\omega_n$. Its density has a particularly simple Laplace transform $\mathbb{E}e^{-\theta V_n} = \left(1 + (\omega_n \theta)^{\lambda_1/\delta_n}\right)^{-1}$. The parameter $\omega_n$ may be computed by the following recurrence relations:

setting $\omega_1 = \alpha_1/\lambda_1$, then for $n \geq 1$,

$$\omega_{n+1} = \begin{cases} \dfrac{v_n}{\delta_n - \lambda_{n+1}} \omega_n & \delta_n > \lambda_{n+1} \quad \text{'stay below max fitness'} \\[2ex] \dfrac{v_n}{r_n} \omega_n & \delta_n = \lambda_{n+1} \quad \text{'equal to max fitness'} \\[2ex] [C_n v_n (\log v_n^{-1})^{r_n - 1} \omega_n]^{\lambda_{n+1}/\delta_n} & \delta_n < \lambda_{n+1} \quad \text{'increase max fitness'} \end{cases} \quad (2)$$

where $c_n = \pi \left(\frac{\alpha_{n+1}}{\lambda_{n+1}}\right)^{\delta_n/\lambda_{n+1}} \left(\alpha_{n+1} \delta_n^{r_n - 1} \sin \frac{\pi \delta_n}{\lambda_{n+1}}\right)^{-1}$. Notably, when type 1 has the maximal growth rate of all types up to type $n$, that is $\delta_n = \lambda_1$, the Mittag-Leffler distribution collapses to an exponential distribution with mean $\omega_n$. Stochastic simulations of the scaled number of type $n$ cells for large times, $e^{-\delta_n t} t^{-(r_n - 1)} Z_n(t) \approx V_n$, which according to Eq (1) is Mittag-Leffler distributed, are compared with theory in Fig 3.

The variable $V_n/\omega_n$ is a single parameter Mittag-Leffler random variable with scale parameter one, and tail parameter $\gamma = \lambda_1/\delta_n$. For $\gamma = 1$ its density is simply $e^{-x}$, and hence $V_n/\omega_n$ has mean 1, while for $\gamma < 1$ the density has a $x^{\gamma - 1}$ singularity at the origin and a $x^{-\gamma - 1}$ tail, thus

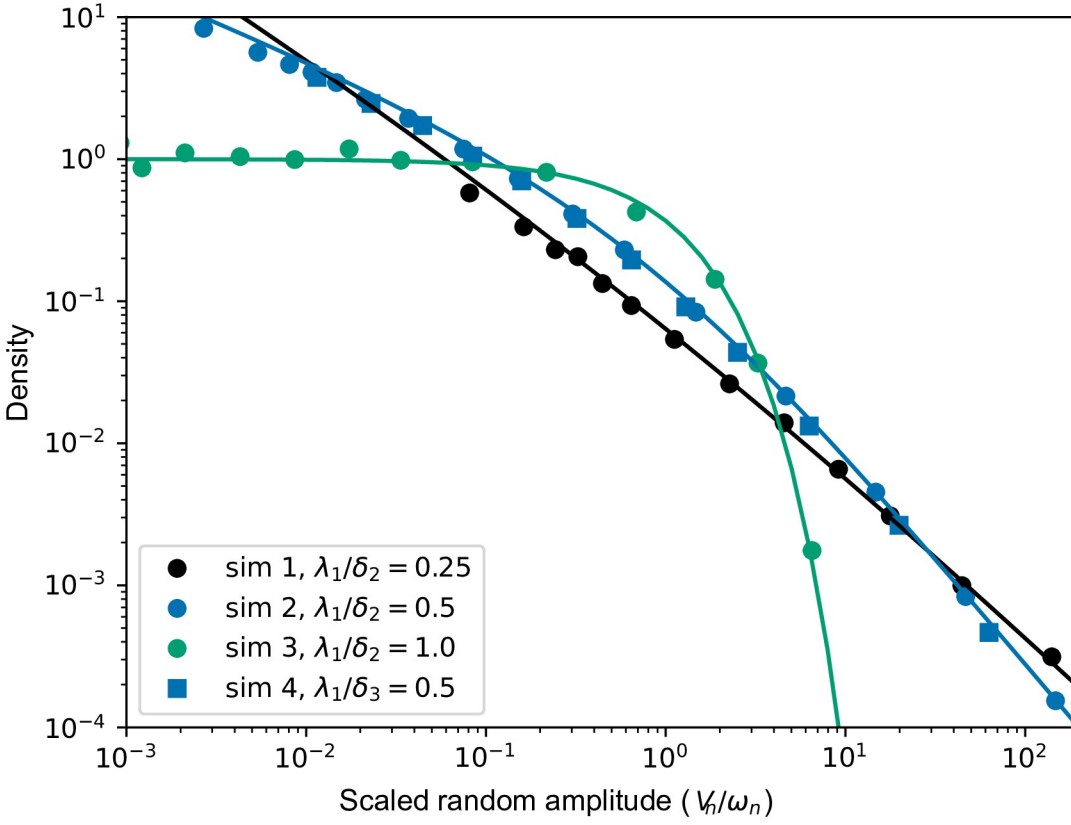

**Fig 3. Comparison of limiting Mittag-Leffler distribution for the number of type $n$ cells with stochastic simulations.** Eq (1), states that for large times and small mutation rates, the scaled number of type $n$ cells, $e^{-\delta_n t} t^{-(r_n - 1)} Z_n(t) \approx V_n$, is approximately Mittag-Leffler distributed with scale $\omega_n$ and tail $\lambda_1/\delta_n$. Here, we compare simulations of the scaled number of type $n$ divided by $\omega_n$, to the density of $V_n/\omega_n$ which is Mittag-Leffler with scale parameter 1, and tail parameter $\lambda_1/\delta_n \in (0, 1]$. We chose three tail parameter values $\lambda_1/\delta_n = 0.25, 0.5, 1.0$, and these curves are depicted with solid lines. The simulation parameter were always $\alpha_1 = 1.2$, $\beta_1 = 0.2$, $v_1 = 0.01$, $\beta_2 = 0.3$ and for $n = 2$ types sim 1: $\alpha_2 = 4.3$, $t = 5$; sim 2: $\alpha_2 = 2.3$, $t = 7$; sim 3: $\alpha_2 = 1.0$, $t = 12$. Then for $n = 3$ types sim 4: as in sim 3 plus $\alpha_3 = 2.4$, $\beta_3 = 0.4$, $v_3 = 0.001$, $t = 12$. Density lines were created in Mathematica using $x^{\gamma - 1} \text{MittagLefflerE}[\gamma, \gamma, -x^\gamma]$.

$V_n/\omega_n$ has infinite mean. A further property is that, when the running-max fitness does not increase between $n$ and $n + 1$, the random variables $V_n$ and $V_{n+1}$ are equal up to a constant factor (perfectly correlated), i.e. with probability 1

$$
V_{n+1} = \begin{cases} \dfrac{v_n}{\delta_n - \lambda_{n+1}} V_n & \delta_n > \lambda_{n+1}, \\[2ex] \dfrac{v_n}{r_n} V_n & \delta_n = \lambda_{n+1}. \end{cases}
\tag{3}
$$

However, in the case $\delta_n < \lambda_{n+1}$, such simple rules do not apply.

In general, the equation for asymptotic growth (1) together with the formulas for $\omega_n$ in (2) enables us to easily answer questions about the population of different cell types. One might ask, for example, whether the number of cells of type $n$ is greater than a given size $k$ and how the growth rates and mutation rates in the system influence this; this problem can be approached using

$$
\mathbb{P}(Z_n(t) > k) \approx \mathbb{P}(V_n > kt^{1-r_n}e^{-\delta_n t}).
$$

Numerically evaluating the resulting distribution function is standard in scientific software (e.g. using the Mittag-Leffler package in R [17]).

**Arrival times.** Similarly to the population sizes, the exact distribution of the arrival time is analytically intractable outside of the simplest settings. For example, the exact probability that type 3 cells arrive by time $t$ is given in Ref. [18] and requires the evaluation of 4 hypergeometric functions. However, when the mutation rates are small simplicity again emerges; the time until the appearance of the first type $n + 1$ cell, $\tau_{n+1}$, has approximately a logistic distribution

$$
\mathbb{P}(\tau_{n+1} > t) \approx \left[1 + \exp(\lambda_1(t - t_{1/2}^{(n+1)}))\right]^{-1}
\tag{4}
$$

with scale given by $\lambda_1^{-1}$ and median given by

$$
t_{1/2}^{(n+1)} = \frac{1}{\delta_n} \log \frac{\delta_n}{\omega_n v_n [\delta_n^{-1}\log(v_n^{-1})]^{r_n-1}}
\tag{5}
$$

where $\omega_n$ is the scale parameter defined in (2). Comparisons of the limiting logistic distribution with simulations are shown in Fig 4, with further simulations provided in the supplementary figure S1 Fig. The population initiated by the first cell of type $n + 1$ could go extinct, and so we might wish to instead consider the waiting time until the first type $n + 1$ cell whose lineage survives. All lineages of type $n + 1$ will eventually go extinct unless $\lambda_{n+1} > 0$. If $\lambda_{n+1} > 0$ then the results given above hold also for the arrival time of the first surviving lineage if we replace $v_n$ by $v_n\lambda_{n+1}/\alpha_{n+1}$.

For the case where each running-max fitness is attained only by one type ($r_i = 1$ for each $i$) then the medians satisfy the following recursion: with

$$
t_{1/2}^{(2)} = \frac{1}{\lambda_1} \log \frac{\lambda_1^2}{\alpha_1 v_1},
\tag{6}
$$

then for $n \geq 2$

$$
t_{1/2}^{(n+1)} = t_{1/2}^{(n)} + \begin{cases} \dfrac{1}{\delta_n} \log \dfrac{\delta_n - \lambda_n}{v_n} & \delta_{n-1} > \lambda_n \\[2ex] \dfrac{1}{\delta_n} \log \dfrac{\delta_n}{v_n} - \dfrac{1}{\delta_{n-1}} \log(c_{n-1}\delta_{n-1}) & \delta_{n-1} < \lambda_n, \end{cases}
\tag{7}
$$

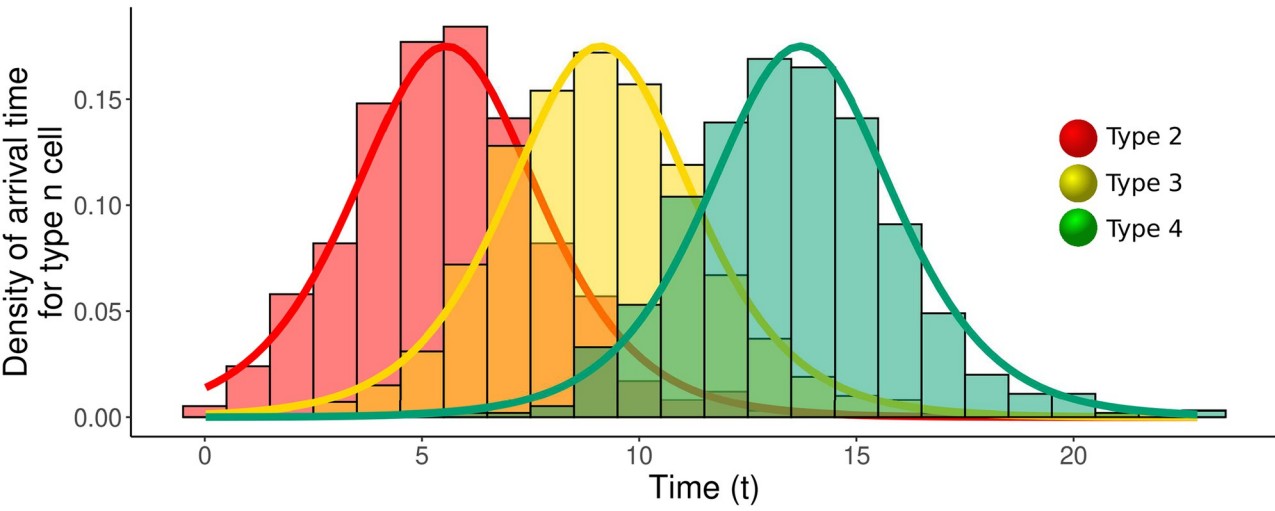

**Fig 4. Comparison of limiting logistic distribution for arrival times with stochastic simulations.** Normalized histogram for the arrival times of types 1–3 obtained from 1000 simulations of the exact model versus the probability density corresponding to the logistic distribution of Eq (4). Note the shape of the distribution remains unchanged. Parameters: $\alpha_1 = \alpha_3 = 1$, $\alpha_2 = 1.4$, $\nu_1 = \nu_2 = \nu_3 = 0.01$, $\beta_1 = \beta_2 = 0.3$, $\beta_3 = 1.5$.

where $c_n$ is defined immediately after Eq (2). If the running-max fitness may be obtained multiple times, then a more detailed recursion also exists, given as Lemma 6 in Methods. Note that since the distribution in Eq (4) is symmetric, the median and the mean coincide.

## Properties of the results

**Population sizes.** From Eq (1), we see that on a logarithmic scale (as in Fig 1C), at large times the number of cells approximately follows a straight line with gradient that increases only when the running-max fitness increases. When the running-max fitness does increase ($\delta_{n-1} < \lambda_n$), then the type $n$ cell number grows exponentially with rate $\lambda_n$. Conversely, if the type $n$ cells have net growth rate smaller than the running-max fitness ($\delta_{n-1} > \lambda_n$), then as the large time behaviour of the type $n$ cell number is exponential growth with rate $\delta_{n-1} = \delta_n$, the flux from the type $n-1$ population eventually drives the cell growth. One can observe this behaviour in Fig 1C: although the type 2 cells have lower fitness than type 1, the population sizes both eventually grow at the same rate of $\lambda_1$. However, the type 3 cells have the largest fitness so far, hence the cell number grows at its own rate $\lambda_3$. When the type $n$ cells have net growth rate equal to the running-max fitness ($\delta_{n-1} = \lambda_n$), relevant for a neutral mutations scenario, then exponential growth at rate $\delta_n$ occurs but with an additional geometric factor of $t^{r_n-1}$. The origin of this geometric factor is best understood by considering the mean growth for $n = 2$, $\lambda_1 = \lambda_2$ [19]. In this case mutations occur at rate proportional to $e^{\lambda_1 s}$ and the average number of descendants from a mutation which occurs at time $s$ is $e^{\lambda_1(t-s)}$ by time $t$. Hence, at time $t$, the mean number of mutants is $\propto \int_0^t e^{\lambda_1 s} e^{\lambda_1(t-s)} ds = te^{\lambda_1 t}$, which is the same geometric factor that appeared as for the limit result Eq (1). Extending this argument to type $n$ explains the geometric factor.

The random amplitude of the deterministic growth, $V_n$, has a Mittag-Leffler distribution, with infinite mean if $\lambda_1 < \delta_n$, which is driven by a power-law decay in its distribution. Intuition for the tails can be gleaned from the case of $n = 2$ [19]. In the $\lambda_1 < \lambda_2$ case, the power-law tail arises due to rare, early mutations from the type 1 cells. The descendants of these early mutations make a considerable contribution to the total number of type 2 cells even at large times

(see discussion of Theorem 3.2 in [19]). However, for $\lambda_1 \geq \lambda_2$, the type 2 descendants from any given mutation eventually make up zero proportion of the type 2 population. Instead, the sheer number of new mutations from the type 1 cells drives the growth of the type 2 population, and in this case the tail decays exponentially. To move to type $n$, from Eq (3) we see that if $\delta_n \geq \lambda_{n+1}$ then the randomness in the cell number is inherited from type $n$ to type $n + 1$. Thus if the running-max fitness does not exceed the growth rate of the type 1 population, that is if $\delta_1 = \delta_n$, then an exponential distribution will be propagated, i.e. all $(V_i)_{i=1}^n$ follow an exponential distribution. However, if the running-max fitness does increase, then for the first $i$ such that $\delta_i < \lambda_{i+1}$, a power-law tail will emerge for $V_{i+1}$. For types that occur after the emergence of the power-law, that is for $j > i + 1$, if the running-max fitness does not increase then the power-law with tail-exponent $\lambda_1/\lambda_{i+1}$ will be propagated, again due to the inheritance property of Eq (3). If instead the running-max fitness increases again, i.e. there is $j > i + 1$ such that $\lambda_{i+1} < \lambda_j$, then the power-law tail remains but with the exponent decreased to $\lambda_1/\lambda_j$. Thus, if the running-max fitness ever rises above $\delta_1$, the tail of the random amplitude has a power-law decay with a monotone decreasing exponent $\lambda_1/\delta_n$.

Our approximation (1) for the cell number of the type $n$ cells is valid for large times. Additionally, small mutation rates are required when the running-max fitness increases, so $\lambda_1 < \delta_n$. Heuristically, we expect the approximation to be valid at large enough times such that the type $n$ cells have been seeded with high probability, that is for $t \gg t_{1/2}^{(n)}$. Around the arrival time for the type $n$ cells, $t \approx t_{1/2}^{(n)}$, fluctuations in the cell number can be greater, which can be seen even in the two-type setting. In the two-type neutral case ($\lambda_1 = \lambda_2$), from Eq (1) we expect that, for $t \gg t_{1/2}^{(2)}$, $Z_2(t) \approx V_2 t e^{\lambda_1 t}$ where $V_2$ is exponentially distributed, and therefore has an exponentially decaying tail. However, for $t \approx t_{1/2}^{(2)}$ (or $e^{\lambda_1 t} \approx v_1^{-1}$), it is known that $Z_2(t)$ has a heavy-tailed distribution, commonly known as the Luria-Delbrück distribution [19–21]. On the other hand, for $\lambda_1 < \lambda_2$, we found that $V_2$ does have a power-law heavy-tail as for the Luria-Delbrück distribution. Therefore, at times around the arrival time for type $n$ cells, the fluctuations in cell number may exceed the characterisation given in Eq (1), but at larger times they are described by the Mittag-Leffler random variable $V_n$. We also note that, in the scale parameter recursion of Eq 2, when mutations are mildly deleterious ($0 < \delta_n - \lambda_{n+1} \ll 1$), the scale parameter can take large values. Therefore, caution should be adopted when using our approximation in this case.

**Arrival times.** The arrival time density has a general shape centred at $t_{1/2}^{(n)}$ (Fig 4). As expected, the median arrival time increases with $n$ or as the mutation rates decreases, and the recursion of Eq 7 explicitly details how these parameters interact. In contrast, the variance of the arrival time is always $\approx \pi^2/(3\lambda_1^2)$. Moreover, the entire shape of the distribution, which is centered around $t_{1/2}^{(n)}$, is determined only by $\lambda_1$. Thus due to the constant variance, for $t_{1/2}^{(n+1)} \gg \pi^2/(3\lambda_1^2)$, modellers may safely ignore the stochastic nature of waiting times and treat the arrival time of the type $n$ cells as deterministic. However, our result raises questions for statistical identifiability; aiming to distinguish between models, e.g. does a phenotype of interest require 2 or 3 mutations, based on fluctuations may be difficult due to the common logistic distribution.

The formulas for the arrival times (7) are valid for small mutation rates, and to leading order the increase in the median arrival time for each new type (i.e. $t_{1/2}^{(n+1)} - t_{1/2}^{(n)}$) is $\delta_n^{-1} \log(v_n^{-1})$. An intuitive understanding can be gained by assuming that: (i) the arrival time for the type $n + 1$ cells approximately occurs when the type $n$ population size reaches $1/v_n$ and (ii) we can ignore fluctuations in population size such that the type $n$ population grows

exponentially as in the deterministic factor of Eq (1). Then, for the case $n = 1$, we simply find $t_{1/2}^{(2)}$ as the time it takes an exponentially growing population to grow from one cell to $1/v_1$, that is we solve $e^{\lambda_1 t_{1/2}^{(2)}} = 1/v_1$, which reproduces the leading order of Eq (6) as $v_1 \to 0$. Similarly, for the arrival times for type $n + 1$, suppose we start an exponential function at $t_{1/2}^{(n)}$ with net growth rate $\delta_n$; this growth will take $\delta_n^{-1} \log(v_n^{-1})$ time to reach the threshold of $v_n^{-1}$ from one cell. To leading order in small mutation rates, this reproduces the recursion of Eq (7).

**Comparison with prior special cases.** Special cases of our results have been obtained previously. Durrett and Moseley [10] obtained the formulas for the arrival time in the special case $\lambda_1 < \lambda_2 < \cdots < \lambda_n$ in the context of accumulation of driver mutations in cancer, and the leading order was also derived in [5]. A key conclusion of [5, 10] follows directly from the representation of the difference in median arrival times given in Eq (7): Assuming a constant driver mutation rate ($v_1 = \ldots = v_n$), the median waiting time between the $n$th and $(n + 1)$th driver mutation is approximately

$$t_{1/2}^{(n+1)} - t_{1/2}^{(n)} = \frac{1}{\lambda_n} \log \frac{\lambda_n}{v_n} - \frac{1}{\lambda_{n-1}} \log c_{n-1}\lambda_{n-1}$$

which decreases as a function of $n$. Hence, under this model, tumor evolution accelerates during its growth [5, 10]. For a comparison with the formulas of [10], note that in this case the running-max fitness for type $j$ is always $\lambda_j$, that is $\delta_j = \lambda_j$, and so $r_j = 1$ for all $j$. Further, the cell types in [10] are numbered from zero. Then the quantity $\omega_{n+1}^{\lambda_1/\lambda_{n+1}}$ as defined in this paper corresponds and agrees with $c_{\theta,n}\mu_n$ of [10] (the formulas in [10] contain some misprints, but they are corrected in [22]). Durrett and Moseley [10] also pointed out that the shapes of the distributions of both the arrival time and the population size were independent of $n$. These distributions were also observed for the special case $\lambda_1 > \lambda_i$ for $1 < i \le n$ in [11]; this case was studied under the motivation of mutations that confer drug resistance but at a fitness cost. In the present paper we have found that even for a general sequence of net growth rates the distribution shapes remain independent of $n$ and their dependence on the rate parameters can be written in relatively simple terms.

## An application: *n*-mutation fluctuation assays

Pairing mathematical models for the emergence of drug resistance during exponential population growth with experimental fluctuation assays enables the inference of mutation rates [1, 23]. In the classic fluctuation assay, replicates are initiated by a small number of drug sensitive cells, which are then grown for either a fixed time period or until the total population reaches a given size. The cells are then exposed to the drug, killing non-resistant cells, which allows the number of replicates without resistance, and the mutant number in those replicates with resistance, to be measured. These experimental quantities are then combined with an appropriate statistical model to infer the mutation rate of acquiring resistance [24]. Originally, only wild type and mutated cells were considered in fluctuation assays. However, including multiple types is required when assessing multidrug resistance, investigating resistant-intermediates such as persistor cells [25], or if multiple gene amplifications are needed for therapy resistance. Gene amplifications are a prevalent resistance mechanism in cancer [26] and amplification rates have been previously reported using fluctuation assays [27], under the standard assumption of a single mutational transition to resistance. However, the modelling assumption of a single mutation imbuing therapy tolerance may be invalid if multiple amplifications are required for resistance. For example, the drug resistant $WB_{20}$ rat epithelial cell line in Tlsty *et al* [27] contained 4 gene copies, compared to the wild type having only 1 copy of the

resistance gene. In such settings, to meaningfully infer amplification rates, an inference framework that describes sequential mutation acquisition is needed. With our results such a modified inference scheme can be constructed.

For simplicity, and as is typical for mutation rate inference, assume mutations are modelled as neutral ($\lambda_1 = \lambda_2 = \ldots$) and that mutations occur at rate $\nu$ ($\nu = \nu_1 = \nu_2 = \ldots$). Suppose $k$ replicates of a fluctuation assay are performed and the number of replicates without resistance, and/or the distribution of mutant numbers over replicates is recorded (Fig 5A). If the mutation rate $\nu$ is known, the distribution of replicates without resistance is binomial with $k$ trials and success probability given by the logistic distribution of Eq (4) (further details on inference methodology is given in the supplementary material S1 Text). In this setting the median arrival time of the $(n + 1)$th type is

$$t_{1/2}^{(n+1)} = \frac{1}{\lambda_1} \log \frac{\lambda_1^2 (n-1)!}{\alpha_1 \left[ \lambda_1^{-1} \log(\nu^{-1}) \right]^{n-1} \nu^n}.$$

Hence, given the number of replicates without resistance, the unknown mutation rate $\nu$ may be inferred by maximum likelihood ($p_0$ method). Similarly, the mutant count distribution over replicates would be characterised by Eq (1), which in this setting take the simple form of

$$Z_n(t) \approx V_n t^{n-1} e^{\lambda_1 t},$$

with $V_n$ an exponential random variable with mean $\omega_n = \frac{\alpha_1}{\lambda_1} \frac{\nu^{n-1}}{(n-1)!}$. Maximum likelihood for the mutant counts under this distribution provides a secondary approach to infer $\nu$.

Fig 5B shows likelihood inference for the mutation rate using both approaches assuming 100 simulated replicates and that 2 mutations (e.g. amplifications) confer resistance. The two inference approaches have strengths and weaknesses depending on the underlying mutation rate and the time $t$ for which the cells are grown before being exposed to the drug. If $t$ is too large ($t \gg t_{1/2}^{(n)}$) the majority, or all, replicates will have resistant cells, and hence the number without resistance carries limited information on the mutation rate (e.g. the wide error bars for $\log_{10}(\nu) = -1.5$ in the left plot of Fig 5B). Instead, the long-time limit approximation of the mutant count distribution, Eq (1), is appropriate, and here our simulated inference for the mutation rate closely matches the true parameter value (Fig 4B). However, if $t$ isn't large enough ($t \approx t_{1/2}^{(n)}$) then Eq (1) poorly characterises the distribution of resistant cells (e.g. the incorrect inference for $\log_{10}(\nu) = -3$ in the right plot of Fig 5B); instead, the $p_0$ method enables accurate inference of the mutation rate. Hence, similar to the advice for the classic fluctuation assay [24], if only some replicates show resistance the $p_0$ method is preferred, whereas if all replicates have sizeable mutant numbers, inference using the mutant counts is advisable. Note that our inference here has assumed known birth rates and no death. These rates could be measured by standard experimental protocols, for example using growth curve assays. Kimmel and Axelrod [28] also gave statistical consideration to a fluctuation assay where two mutations are needed. However, in principle (neglecting experimental complexities), our results hold for any $n$, include death, and allow for varied growth rates between the cell types, extending the work of Ref. [28].

## Discussion

Due to their simplicity and ability to model fundamental biology such as cell division, death, and mutation, multitype branching processes have become a standard tool for quantitative researchers investigating evolutionary dynamics in exponentially growing populations. Further, these models are able to link detailed microscopic molecular processes to explain

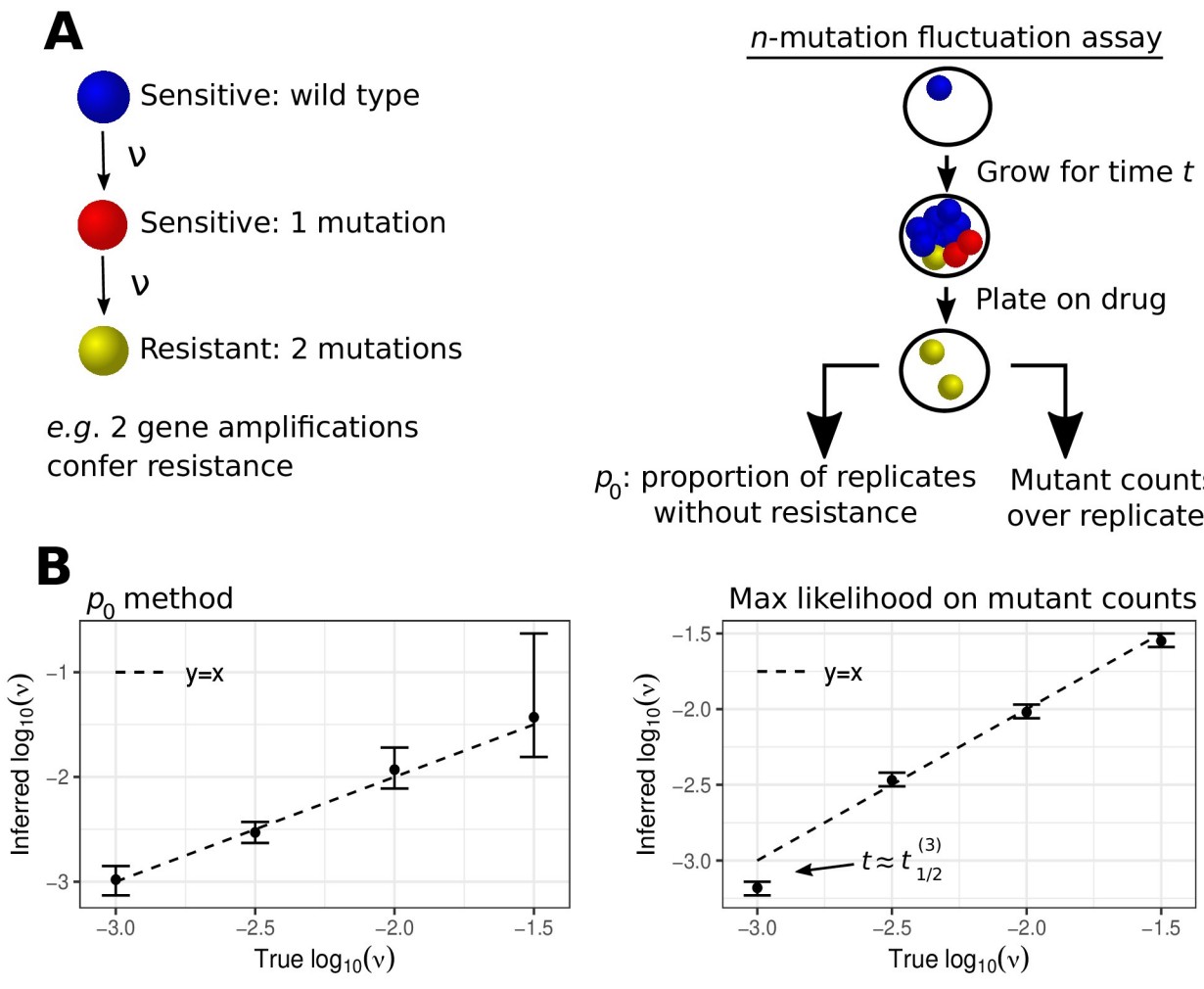

**Fig 5. Statistical inference for an *n*-mutation fluctuation assay.** A. Schematic of a fluctuation assay for the measurement of mutation rates when *n* mutations are required for resistance. Drug sensitive cells are initially cultured, and after growth for a given time *t*, the cells are exposed to a selective medium. Non-resistant cells are killed, revealing the number of mutants. This experiment is conducted over replicates, and the number of replicates without resistance and the mutant numbers are recorded. B. Likelihood inference on a simulated fluctuation assay assuming: 2 mutations are required for resistance, 100 replicates, no death, $\alpha_i = 1$ for each *i*, $t = 10$, and the mutation rate *ν* stated on the *x*-axis. Wide error bars are expected when using the $p_0$ method for $t \gg t_{1/2}^{(n)}$ as only a small number of replicates have no resistant cells; in such a setting using the mutant counts (right panel) provides superior inference. Likewise, if $t \approx t_{1/2}^{(n)}$ the approximation of Eq (1) is not appropriate, which explains the inaccurate inference for $\log_{10}(\nu) = -3$ when using the mutant counts; the $p_0$ method provides improved inference in this scenario.

macroscopic experimental, clinical, and epidemiological data [29, 30]. Despite the importance of this framework, even simple questions are often challenging to examine. Whilst numerical and simulation based methods have proven powerful for both model exploration and statistical inference, the computational expense of simulating to plausible scales can lead to challenges; e.g. simulating to tumour sizes orders of magnitude smaller than reality, which provides obstacles for biological interpretation of inferred parameters. Moreover, it is often unclear how to precisely summarise the manner in which a large number of parameters interact to influence quantities of interest, such as the time until a triply resistant cell emerges. In this study, we analysed the regimes of large times, and small mutation rates, in order to develop limiting formulas that can be used to quickly gain intuition or for approximate statistical inference

We have focused on the number, and arrival time, of cells with *n* mutations. While this problem dates back at least to the work of Luria and Delbrück—where a mutation resulted in

phage resistant bacteria—specific instances of the problem are commonly used to study a variety of biological phenomena [3–5, 8, 9, 14, 24, 31–34]. The time of first mutation is well known, however the arrival time of cells with $n$ alterations is unclear outside of specific fitness landscapes [10, 11]. Here, we developed approximations for the cell number and arrival time regardless of whether mutations increase, decrease, or have no effect on the growth rate of the cells carrying the alterations. We showed that, within relevant limiting regimes, the number of type $n$ cells can be decoupled into the product of a deterministic time-dependent function and a time-independent Mittag-Leffler random variable; meanwhile the arrival time of type $n$ cells follows a logistic distribution with a shape that depends only on the net growth of the type 1 cells. The features of these distributions, such as median arrival time, can be exactly mapped to the underlying model parameters, that is the division, death, and mutation rates. These results illuminate the effects of mutation and selection, and can be readily numerically evaluated to explore particular biological hypotheses. We highlighted the utility of our results on mutation rate inference in fluctuation assays.

As the biological processes studied become increasingly complex, so too will the mathematical models constructed to describe such processes. We hope that the results of the present paper will enable researchers to find simplicity in an arbitrarily complex parameter landscape for a fundamental class of mathematical models.

## Methods

In this section we provide detailed results and proofs in their general form.

### Branching process: Population growth

We first look to understand the number of cells of type $n$ at time $t$, that is $Z_n(t)$, at large times.

**Proposition 1.** Assume non-extinction of the type 1 population, that is that $Z_1(t) > 0$ for all $t \geq 0$. Then, for each $n \in \mathbb{N}$, there exists a $(0, \infty)$-valued random variable $V_n$ such that

$$\lim_{t \to \infty} t^{-r_n+1} e^{-\delta_n t} Z_n(t) = V_n$$

almost surely.

As our branching process is reducible this result is not considered classical [35]. Heuristically, the result says that for large $t$, $Z_n(t) \approx V_n t^{r_n-1} e^{\delta_n t}$ and so at large times all the stochasticity of $Z_n(t)$ is bundled into the variable $V_n$.

Towards proving Proposition 1, we first consider a model of a deterministically growing population which seeds mutants as a Poisson process, the mutants growing as a branching process. The next result defines the model and describes the large-time number of mutants, generalising a result of [36].

**Lemma 1.** *Let $(f(t))_{t \geq 0}$ be a non-negative cadlag function, $x, \delta > 0$, and $r \geq 0$, with*

$$\lim_{t \to \infty} t^{-r} e^{-\delta t} f(t) = x.$$

*Suppose that $(T_i)_{i \in \mathbb{N}}$ come from a Poisson process on $[0, \infty)$ with intensity $f(\cdot)$. Suppose that $(Y_i(t))_{t \geq 0}$, $i \in \mathbb{N}$, are i.i.d. birth-death branching processes initiating from a single cell, that is $Y_i(0) = 1$, with birth and death rates $\alpha$ and $\beta$. Let $\lambda = \alpha - \beta$. Define*

$$Z(t) = \sum_{i : T_i \leq t} Y_i(t - T_i).$$

*Then*

$$
\begin{cases}
\lim_{t \to \infty} t^{-r} e^{-\delta t} Z(t) = \dfrac{x}{\delta - \lambda}, & \text{for } \delta > \lambda; \\[2mm]
\lim_{t \to \infty} t^{-r-1} e^{-\delta t} Z(t) = \dfrac{x}{r+1}, & \text{for } \delta = \lambda; \\[2mm]
\lim_{t \to \infty} e^{-\lambda t} Z(t) = V, & \text{for } \delta < \lambda;
\end{cases}
$$

*almost surely. Here V is some positive random variable with mean $\int_0^\infty e^{-\lambda s} f(s) ds$.*

*Proof.* We first give the argument assuming $\lambda \neq 0$, and provide a comment at the end of the proof indicating modifications needed for the $\lambda = 0$ case.

First we claim that

$$
M(t) = e^{-\lambda t} Z(t) - \int_0^t e^{-\lambda s} f(s) ds, \quad t \geq 0;
$$

is a martingale with respect to the natural filtration. Indeed, for $s \leq t$,

$$
\begin{aligned}
\mathbb{E}[M(t)|\mathcal{F}_s] &= e^{-\lambda t} \mathbb{E}[Z(t)|\mathcal{F}_s] - \int_0^t e^{-\lambda u} f(u) du \\
&= e^{-\lambda t}\left(Z(s) e^{\lambda(t-s)} + \int_s^t f(u) e^{\lambda(t-u)} du\right) - \int_0^t e^{-\lambda u} f(u) du \\
&= M(s),
\end{aligned}
$$

as required.

Next we look to bound the second moment of $M(t)$. To this end, observe that $Z(t) = \sum_{i: T_i \leq t} Y_i(t - T_i)$ is a compound Poisson distribution which is a Poisson $\left(\int_0^t f(s) ds\right)$ sum of i.i.d. random variables distributed as $Y_1(t - \xi)$, where $\xi$ is a $[0, t]$-valued random variable with density proportional to $f$ (see, e.g., Section 2 of [36]). Using the already-known second moment for a birth-death branching process [37] (see Theorem 6.1 on page 103),

$$
\mathbb{E}\left[Y_i(t)^2\right] = \frac{2\alpha}{\lambda} e^{2\lambda t} - \frac{\alpha + \beta}{\lambda} e^{\lambda t},
$$

we have that

$$
\mathbb{E}\left[Y_1(t - \xi)^2\right] = \frac{\int_0^t f(s)\left(\frac{2\alpha}{\lambda} e^{2\lambda(t-s)} - \frac{\alpha+\beta}{\lambda} e^{\lambda(t-s)}\right) ds}{\int_0^t f(s) ds}.
$$

It follows that

$$
\begin{aligned}
\mathrm{Var} Z(t) &= \mathbb{E}[Y_1(t - \xi)^2] \int_0^t f(s) ds \\
&= \int_0^t f(s)\left(\frac{2\alpha}{\lambda} e^{2\lambda(t-s)} - \frac{\alpha+\beta}{\lambda} e^{\lambda(t-s)}\right) ds,
\end{aligned}
$$

and since $\mathbb{E}M(t) = 0$, we find that

$$
\begin{aligned}
\mathbb{E}[M(t)^2] &= \text{Var}[M(t)] = e^{-2\lambda t}\text{Var}Z(t) \\
&= e^{-2\lambda t}\int_0^t f(s)\left(\frac{2\alpha}{\lambda}e^{2\lambda(t-s)} - \frac{\alpha+\beta}{\lambda}e^{\lambda(t-s)}\right)ds \\
&\leq \frac{2\alpha}{\lambda}\int_0^t e^{-2\lambda s}f(s)ds \\
&= \frac{2\alpha}{\lambda}\int_0^t (s\vee 1)^r e^{(\delta-2\lambda)s}(s\vee 1)^{-r}e^{-\delta s}f(s)ds \\
&\leq \frac{2\alpha}{\lambda}\sup_{s\geq 0}\left[(s\vee 1)^{-r}e^{-\delta s}f(s)\right]\int_0^t (s\vee 1)^r e^{(\delta-2\lambda)s}ds.
\end{aligned}
$$

Therefore

$$
\mathbb{E}[M(t)^2] \leq
\begin{cases}
Ct^r e^{(\delta-2\lambda)t}, & \text{for } \delta > 2\lambda; \\
Dt^{r+1}, & \text{for } \delta = 2\lambda; \\
E, & \text{for } \delta < 2\lambda,
\end{cases}
\tag{8}
$$

where $C$, $D$ and $E$ are positive constants.

To conclude the proof, we will separately consider the three cases listed in the Lemma's statement: $\delta < \lambda$, $\delta = \lambda$, and $\delta > \lambda$.

We begin with the case $\delta < \lambda$. Here the martingale $M(t)$ has a bounded second moment. By the martingale convergence theorem, $M(t)$ converges to some random variable $V'$ with mean zero. Rearranging the limit of $M(t)$,

$$
\lim_{t\to\infty} e^{-\lambda t}Z(t) = \int_0^\infty e^{-\lambda s}f(s)ds + V' =: V,
$$

almost surely, where the integral converges because the integrand has an exponentially decaying tail. The positivity of $V$ can be seen by Fatou's lemma:

$$
\begin{aligned}
\lim_{t\to\infty} e^{-\lambda t}Z(t) &\geq \liminf_{t\to\infty} e^{-\lambda t}Z(t) \\
&\geq \sum_{i\geq 1} e^{-\lambda T_i}\liminf_{t\to\infty} 1_{\{T_i<t\}}e^{-\lambda(t-T_i)}Y_i(t-T_i) \qquad \text{(Fatou's lemma)} \\
&= \sum_{i\geq 1} e^{-\lambda T_i}W_i
\end{aligned}
$$

where the $W_i = \liminf_{t\to\infty} 1_{\{T_i<t\}}e^{-\lambda(t-T_i)}Y_i(t-T_i)$ are i.i.d. random variables on $[0,\infty)$ that are each non-zero with positive probability [22, 35] (recall this case assumes that $\lambda > \delta > 0$ so that each $Y_i(\cdot)$ is supercritical). Hence, with probability one at least one of the $W_i$ is positive. This gives the result for $\delta < \lambda$.

The second case is $\delta = \lambda$. Here the second moment of $M(t)$ is still bounded and so we can again apply the martingale convergence theorem to see that $M(t)$ converges almost surely. It follows that

$$
t^{-r-1}M(t) = t^{-r-1}e^{-\delta t}Z(t) - t^{-r-1}\int_0^t e^{-\delta s}f(s)ds
$$

converges to zero almost surely. Thus, using dominated convergence,

$$
\begin{aligned}
\lim_{t\to\infty} t^{-r-1} \int_0^t e^{-\delta s} f(s) ds &= \lim_{t\to\infty} \int_0^1 u^r (tu)^{-r} e^{-\delta tu} f(tu) du \\
&= x \int_0^1 u^r du \\
&= \frac{x}{r+1}
\end{aligned}
$$

is the almost sure limit of $t^{-r-1} e^{-\delta t} Z(t)$.

The third and final case is $\delta > \lambda$. This case requires a new perspective because the second moment of $M(t)$ may not be bounded, disallowing the martingale convergence theorem. Instead we appeal to Borel-Cantelli. For $\epsilon > 0$ and $n \in \mathbb{N}$, consider the events

$$
B_n^\epsilon := \left\{ \sup_{t \in [n, n+1]} \left( t^{-r} e^{(\lambda-\delta)t} M(t) \right)^2 > \epsilon \right\}.
$$

Then

$$
\begin{aligned}
\mathbb{P}[B_n^\epsilon] &\leq \mathbb{P}\left[ \sup_{t \in [n, n+1]} M(t)^2 > \epsilon n^{2r} e^{2(\delta-\lambda)n} \right] \\
&\leq \frac{\mathbb{E}[M(n+1)^2]}{\epsilon n^{2r} e^{2(\delta-\lambda)n}} \\
&\leq G e^{-\gamma n},
\end{aligned}
$$

by Doob's martingale inequality and then Eq (8); here $G$ and $\gamma$ are positive numbers which do not depend on $n$. By Borel-Cantelli, the probability that only finitely many of $(B_n^\epsilon)_{n \in \mathbb{N}}$ occur is one. Equivalently,

$$
t^{-r} e^{(\lambda-\delta)t} M(t) = t^{-r} e^{-\delta t} Z(t) - t^{-r} e^{(\lambda-\delta)t} \int_0^t e^{-\lambda s} f(s) ds
$$

converges to zero almost surely. Thus, using dominated convergence,

$$
\begin{aligned}
&\lim_{t\to\infty} t^{-r} e^{(\lambda-\delta)t} \int_0^t e^{-\lambda s} f(s) ds \\
&= \lim_{t\to\infty} \int_0^t (t^{-r}(t-s)^r (t-s)^{-r} e^{-\delta(t-s)} f(t-s)) e^{(\lambda-\delta)s} ds \\
&= \int_0^\infty x e^{(\lambda-\delta)s} ds \\
&= \frac{x}{\delta - \lambda}
\end{aligned}
$$

is the almost sure limit of $t^{-r} e^{-\delta t} Z(t)$.

For the case of $\lambda = 0$, minor modifications are required. Firstly, the second-moment has the form

$$
\mathbb{E}[Y_i(t)^2] = 1 + 2\alpha t,
$$

and hence

$$
\begin{aligned}
\mathbb{E}[M(t)^2] = & \int_0^t f(s)(1 + 2\alpha(t - s))\, ds \\
\leq & \int_0^t f(s)(1 + 2\alpha t)\, ds \\
= & (1 + 2\alpha t) \int_0^t (1 \vee s)^r e^{\delta s}(1 \vee s)^{-r} e^{-\delta s} f(s)\, ds \\
\leq & (1 + 2\alpha t)\sup_{s \geq 0}[(1 \vee s)^{-r} e^{-\delta s} f(s)] \int_0^t (1 \vee s)^r e^{\delta s}\, ds \\
\leq & C'(1 + t)e^{\delta t} t^r,
\end{aligned}
$$

with $C'$ a positive constant. When $\lambda = 0$, then $\delta > \lambda$. Thus, the above bound should be used in the Borel-Cantelli centred argument, which leads to the same result.

We can now give the proof of Proposition 1 on the convergence of cell numbers.

*Proof of Proposition 1.* We prove the result by induction. Clearly it is true for $n = 1$. Now suppose that

$$
\lim_{t \to \infty} t^{-(r_n - 1)} e^{-\delta_n t} Z_n(t) = V_n \in (0, \infty)
$$

almost surely. Condition on the trajectory of $Z_n(\cdot)$, and apply Lemma 1 to see that

$$
\lim_{t \to \infty} t^{-(r_{n+1} - 1)} e^{-\delta_{n+1} t} Z_{n+1}(t) = V_{n+1} \in (0, \infty)
$$

almost surely.

Having proven that the cell numbers grow asymptotically as a deterministic function of time multiplied by a time-independent random amplitude $V_n$, our next aim is to determine the distribution of this random amplitude. We shall proceed via induction. To establish the base case we restate a classic result [22, 35]:

**Lemma 2.** *The random variable $V_1$ from Proposition 1 has exponential distribution with parameter $\lambda_1/\alpha_1 = 1 - \beta_1/\alpha_1$.*

Since the type $n$ population seeds the type $n + 1$ population, one might expect that the random amplitudes $V_n$ and $V_{n+1}$ of the two populations are related. The next result says that this is indeed the case for a part of parameter space—when the type $n + 1$ fitness is no greater than the fitnesses of previous types.

**Corollary 1.** *Let $n \geq 1$. If $\delta_n > \lambda_{n+1}$*

$$
V_{n+1} = \frac{v_n V_n}{\delta_n - \lambda_{n+1}} \quad a.s.,
$$

*while for $\delta_n = \lambda_{n+1}$*

$$
V_{n+1} = \frac{v_n V_n}{r_n} \quad a.s.
$$

*Proof.* Immediate from Lemma 1.

Corollary 1 focuses on the case that the fitness of type $n + 1$ does not dominate the fitnesses of types 1 to $n$; here it says that the random amplitude $V_{n+1}$ is simply a constant multiple of $V_n$, meaning that the large-time stochasticity of the type $n + 1$ population size is perfectly inherited from the type $n$ population. A special example is that type 1 has a larger fitness than all subsequent types, in which case $V_n$ is a constant multiple of $V_1$ and thus all random amplitudes are

exponentially distributed, recovering a result of [11]. Corollary 1 is also a generalisation of Theorem 3.2 parts 1 and 2 of [19] which provided the distribution of $V_2$ in terms of $V_1$.

The remaining region of parameter space—where a new type may have a fitness greater than the fitness of all previous types is our next focus. Here, contrasting with the region considered in Corollary 1, the random amplitudes seem to be rather complex. The distribution of $V_2$ takes an intricate form, which is calculated in [16] (Eq. 56) and we do not restate it here for brevity. The distribution of $V_n$ for $n > 2$ apparently are unknown. We aim to find simple approximations for the $V_n$ by introducing an approximate model.

## Approximate model introduction

The exact distribution of the random amplitude $V_n$ for a generic sequence of birth and death rates appears to be analytically intractable. Thus we look to approximate $V_n$ in the limit of small mutation rates. Towards such an approximation, we choose to follow a method inspired by Durrett and Moseley [10] which simplifies calculations by introducing an approximate model. The approximate model is motivated by the following heuristic argument: mutations to create cells of type $(n + 1)$ occur at rate $v_n Z_n(t)$; when the mutation rates are small it will take some time for the first cell of type $(n + 1)$ to appear; at large times $Z_n(t) \sim V_n e^{\delta_n t} t^{r_n - 1}$ (Proposition 1); therefore for small mutation rates, mutations to create cells of type $(n + 1)$ should occur at rate $\approx v_n V_n e^{\delta_n t} t^{r_n - 1}$. We carefully define the approximate model momentarily, but briefly it arises by assuming the type $(n + 1)$ arrive at rate $v_n V_n e^{\delta_n t} t^{r_n - 1}$ and then letting the type $(n + 1)$ cells follow the dynamics we've already been assuming.

Formally, we define the approximate model iteratively. We let $Z_n^*(t)$ be the size of the type $n$ population at time $t$, set $Z_1^*(t) = V_1 e^{\lambda_1 t}$ for $t \geq 0$, and fix $V_1^* = V_1$. Then, given $V_n^*$, let $(T_{n+1,i}^*)$ be the times from a Poisson process with rate

$$t^{r_n - 1} e^{\delta_n t} v_n V_n^*.$$

Then, we set

$$Z_{n+1}^*(t) = \sum_{i : T_{n+1,i}^* \leq t} Y_{n+1,i}\big(t - T_{n+1,i}^*\big) \tag{9}$$

where the $Y_{n,i}(\cdot)$ are independent birth-death processes initiated from a single cell with birth and death rates $\alpha_n$ and $\beta_n$, and

$$V_{n+1}^* = \lim_{t \to \infty} t^{-r_{n+1}+1} e^{-\delta_{n+1} t} Z_{n+1}^*(t). \tag{10}$$

We hypothesise but do not prove that the distribution of the random amplitudes $V_n^*$ and $V_n$ for the approximate and original models respectively coincide in the limit of small mutation rates; this is known to be true in the two-type setting (Section 4.4 of [16]).

## Approximate model: Population growth

First we have the counterpart to Proposition 1, clarifying that the approximate model is well defined.

**Proposition 2.** For $n \geq 1$, there exists a $(0, \infty)$-valued random variable $V_n^*$ such that

$$\lim_{t \to \infty} t^{-(r_n - 1)} e^{-\delta_n t} Z_n^*(t) = V_n^*$$

almost surely.

*Proof.* Identical to the proof of Proposition 1.

Analogously to Corollary 1 we can relate the random amplitudes of type $n + 1$ with that of type $n$ for the approximate process—now we include also the case where type $n$ has a larger growth rate than the type $(n − 1)$ cells. We give the results at the level of the Laplace transform, as it turns out this function will dictate the distribution of the arrival times.

**Corollary 2.** *Let $n \geq 1$. Then*

$$\mathbb{E}[\exp(-\theta V_{n+1}^*)] = \mathbb{E}[\exp(-h_n(\theta) V_n^*)],$$

*where $h_n(\theta)$ is defined by*

$$h_n(\theta) = \begin{cases} \dfrac{v_n \theta}{\delta_n - \lambda_{n+1}} & \delta_n > \lambda_{n+1} \\[2ex] \dfrac{v_n \theta}{r_n} & \delta_n = \lambda_{n+1} \\[2ex] v_n \dfrac{\theta(r_n - 1)!}{\lambda_{n+1}^{r_n}} \Phi(-\theta \alpha_{n+1}/\lambda_{n+1}, r_n, 1 - \delta_n/\lambda_{n+1}) & \delta_n < \lambda_{n+1}, \end{cases}$$

*where $\Phi$ is the Lerch transcendent function (see 25.14.1 in [38]).*

*Proof.* For the cases of $\delta_n > \lambda_{n+1}$ or $\delta_n = \lambda_{n+1}$ we can appeal directly to Corollary 1.

For $\delta_n < \lambda_{n+1}$, we expand upon the argument of Durrett and Moseley [10], who considered $\lambda_1 < \lambda_2 < \dots$. Let $\zeta_{n+1}(t, z) = \mathbb{E}e^{-zY_{n+1,1}(t)}$ which is the Laplace transform for a linear birth-death process initiated with a single cell, at time $t$ with division and death rates $\alpha_n, \beta_n$. Note that when $\delta_n < \lambda_{n+1}$, necessarily $\lambda_{n+1} > 0$ as $\delta_n \geq \delta_1 > 0$, due to the type 1 population being assumed supercritical. If we fix $V_n^*$, then the arrivals to the type $n + 1$ population occur as a Poisson process, so by the definition of $Z_{n+1}^*(t)$ given in Eq (9), $Z_{n+1}^*(t)$ is a compound Poisson random variable. Generally, if we have a compound Poisson variable, defined by the sum of $N \sim \text{Poisson}(\lambda)$ i.i.d. random variables $X_i$, then its Laplace transform follows

$$\mathbb{E}\exp\left(-\theta \sum_{i=1}^N X_i\right) = \exp[-\lambda(1 - \mathbb{E}e^{-\theta X_1})].$$

In our case, with $V_n^*$ fixed, $Z_{n+1}^*(t)$ is a Poisson $(\int_0^t v_n V_n^* s^{r_n - 1} e^{\delta_n s} ds)$ sum of i.i.d. random variables distributed as $Y_1(t - \xi)$, where $\xi$ is a $[0, t]$-valued random variable with density proportional to $v_n V_n^* s^{r_n - 1} e^{\delta_n s}$ (see, e.g., Section 2 of [36]). Applying this to $Z_{n+1}^*(t)$ we have

$$\mathbb{E}[\exp(-e^{-\lambda_{n+1}t} Z_{n+1}^*(t)\theta)|V_n^*] =$$

$$\exp\left(-v_n V_n^* \int_0^t s^{r_n - 1} e^{\delta_n s} [1 - \zeta_{n+1}(t - s, \theta e^{-\lambda_{n+1}t})] ds\right). \tag{11}$$

To obtain the limit of the integrand we use the well known result (see Ref. [10] Section 2) that if $Y(\cdot)$ is a linear birth-death process with division, and death rates $\alpha_{n+1}, \beta_{n+1}$, initiated from a single cell ($Y(0) = 1$), and with $\phi_{n+1} = \lambda_{n+1}/\alpha_{n+1}$, then as $t \to \infty$, $e^{-\lambda_{n+1}t} Y(t) \xrightarrow{d} B \times E$ where $B \sim \text{Bernoulli}(\phi_{n+1})$, $E \sim \text{Expo}(\phi_{n+1})$, and both random variables are independent from each other. Hence its Laplace transform converges to

$$\mathbb{E}\exp(-\theta Y(t)e^{-\lambda_{n+1}t})$$

$$\to 1 - \phi_{n+1} + \phi_{n+1} \int_0^\infty e^{-\theta x} \phi_{n+1} e^{-\phi_{n+1}x} dx = 1 - \phi_{n+1}\left(1 - \frac{1}{1 + \theta/\phi_{n+1}}\right)$$

Then

$$1 - \zeta_{n+1}(t - s, \theta e^{-\lambda_{n+1}t}) = 1 - \mathbb{E}\exp(-\theta e^{-\lambda_{n+1}s}e^{-\lambda_{n+1}(t-s)}Y(t-s))$$

$$\rightarrow \phi_{n+1}\left(1 - \frac{1}{1 + \theta e^{-\lambda_{n+1}s}/\phi_{n+1}}\right)$$

as $t \to \infty$. Using this and taking the $t \to \infty$ limit over Eq 11 results in

$$\lim_{t\to\infty} \mathbb{E}[\exp(-e^{-\lambda_{n+1}t}Z_{n+1}^*(t)\theta)|V_n^*] =$$

$$\exp\left(-v_n V_n^* \phi_{n+1} \int_0^\infty s^{r_n-1}e^{\delta_n s}\left(1 - \frac{1}{1 + e^{-\lambda_{n+1}s}\theta/\phi_{n+1}}\right) ds\right)$$

Let $\gamma_n = \delta_n/\delta_{n+1}$ and recall the Lerch transcendent has integral representation for $\mathcal{R}s > 0$, and $\mathcal{R}a > 0$ (see 25.14.5 in [38])

$$\Phi(z, s, a) = \frac{1}{\Gamma(s)} \int_0^\infty \frac{t^{s-1}e^{-at}}{1 - ze^{-t}} dt$$

which converges for $z \in \mathbb{C} \setminus [1, \infty)$. Upon the substitution $t = \lambda_{n+1}s$ we see

$$
\begin{aligned}
h_n(\theta) &= v_n \phi_{n+1} \int_0^\infty s^{r_n-1}e^{\delta_n s}\left(1 - \frac{1}{1 + e^{-\lambda_{n+1}s}\theta/\phi_{n+1}}\right) ds \\
&= \frac{v_n \theta}{\lambda_{n+1}^{r_n}} \int_0^\infty \frac{t^{r_n-1}e^{-(1-\gamma_n)t}}{1 + \theta e^{-t}/\phi_{n+1}} dt \\
&= \frac{v_n \theta \Gamma(r_n)}{\lambda_{n+1}^{r_n}} \Phi(-\theta/\phi_{n+1}, r_n, 1 - \gamma_n) \\
&= \frac{v_n \theta (r_n - 1)!}{\lambda_{n+1}^{r_n}} \Phi(-\theta/\phi_{n+1}, r_n, 1 - \gamma_n).
\end{aligned}
$$

Corollary 2 implies that

$$
\begin{aligned}
\mathbb{E}[\exp(-V_n^*\theta)] &= \mathbb{E}[\exp(-V_1^* h_1 \circ \ldots \circ h_{n-1}(\theta))] \\
&= (1 + h_1 \circ \ldots \circ h_{n-1}(\theta)\alpha_1/\lambda_1)^{-1},
\end{aligned}
\tag{12}
$$

which means that the distribution of the random amplitude $V_n^*$ is possible to numerically evaluate. Such numerical computation for the approximate model is already a step beyond what we could do for the original model.

Recall that it was heuristically argued that the random amplitudes of the approximate and original models coincide in the limit of small mutation rates. Therefore the exact distribution of $V_n^*$ seen in (12) is not so much our interest as is its limit for small mutation rates. Our task for the remainder of this section is thus to take the small mutation rate limit of (12).

To state the limit we now introduce some notation.

Let

$$
f_i(v_i) = \begin{cases} v_i^{-1} & \lambda_{i+1} \leq \delta_i \\ v_i^{-1}\log(v_i^{-1})^{-(r_i-1)} & \lambda_{i+1} > \delta_i \end{cases}
\tag{13}
$$

Then, writing $\boldsymbol{v} = (v_1, v_2, ..)$, we define

$$\mathcal{F}_n(\boldsymbol{v}) = \prod_{i=1}^{n} f_i(v_i)^{\delta_{n+1}/\delta_i}. \tag{14}$$

This function satisfies

$$\mathcal{F}_n(\boldsymbol{v}) = (f_n(v_n)\mathcal{F}_{n-1}(\boldsymbol{v}))^{\delta_{n+1}/\delta_n} \tag{15}$$

Further let $\gamma_n = \delta_n/\delta_{n+1}$, and

$$\kappa_n = \begin{cases} (\delta_n - \lambda_{n+1})^{-1} & \delta_n > \lambda_{n+1} \\ r_n^{-1} & \delta_n = \lambda_{n+1} \\ \dfrac{\phi_{n+1}^{1-\gamma_n}}{\lambda_{n+1}^{r_n} \gamma_n^{r_n-1}} \dfrac{\pi}{\sin \gamma_n \pi} & \delta_n < \lambda_{n+1}. \end{cases} \tag{16}$$

Note that $c_n$ as defined under Eq (2) is $\kappa_n$ when $\delta_n < \lambda_{n+1}$. Then, for small mutation rates, the distribution of $V_n^*$ may be related to $V_1^*$:

**Proposition 3**.

$$\lim_{v_1 \to 0} \dots \lim_{v_n \to 0} \mathbb{E}[\exp(-V_{n+1}^* \theta \, \mathcal{F}_n(\boldsymbol{v}))] = \mathbb{E}\left[\exp\left(-V_1^* \theta^{\delta_1/\delta_{n+1}} \prod_{i=1}^{n} \kappa_i^{\delta_1/\delta_i}\right)\right]$$

$$= \left(1 + (\alpha_1/\lambda_1)\theta^{\delta_1/\delta_{n+1}} \prod_{i=1}^{n} \kappa_i^{\delta_1/\delta_i}\right)^{-1}$$

Before proving this proposition we give two required lemmas in order to understand the limit behaviour of the function $h_n(\theta)$ (defined in Corollary 2). Recall the Lerch transcendant function appeared in the definition of $h_n(\theta)$, which motivates considering the following lemma.

**Lemma 3.** *With $\Phi$ as the Lerch transcendent function with $0 < a < 1$ and positive integer $s$, as $z \to -\infty$*

$$\Phi(z, s, a) \sim \frac{\pi}{\sin a\pi} \frac{1}{(-z)^a} \frac{(\log -z)^{s-1}}{(s-1)!}.$$

*Proof.* We first rewrite $\Phi$ in terms of the generalised hypergeometric function (see 16.2.1 in [38]) for positive integer $s$

$$\Phi(z, s, a) = a^{-s}{}_{s+1}F_s\left(\begin{matrix} 1, a, \cdots, a \\ a+1, \cdots, a+1 \end{matrix}; z\right).$$

This identity can be readily verified from the definitions of these special functions. Then we use its integral representation (Eq. 16.5.1 at [38])

$$\Phi(z, s, a) = \frac{1}{2\pi i} \int_{-i\infty}^{i\infty} \frac{\Gamma(1+x)\Gamma(-x)}{(a+x)^s} (-z)^x dx$$

The integrand has poles at $-a$ (where $0 < a < 1$) and at all real integers due to the Gamma functions. The contour of integration separates the poles at $-a$ and 0. From the residue theorem for $z < 0$ we can rewrite the integral as the sum of the residues coming from all poles on

the left of the contour

$$\Phi(z, s, a) = \text{Res}_{x=-a}\left(\frac{\Gamma(1+x)\Gamma(-x)}{(a+x)^s}(-z)^x\right) + (-1)^s \sum_{n=1}^{\infty}\frac{z^{-n}}{(n-a)^s}.$$

The first term on the right hand side is the contribution from the pole at $-a$, while the sum goes over the contributions from all other poles at $-n = -1, -2, \ldots$. The leading order term comes from the residue of closest pole to the origin at $x = -a$, which can be written as a finite sum of terms including powers of $\log - z$. The leading order of these terms is

$$\Phi(z, s, a) \sim \frac{\pi}{\sin a\pi}\frac{(\log - z)^{s-1}}{(s-1)!(-z)^a} + O\left(\frac{(\log - z)^{s-2}}{(-z)^a}\right)$$

Before giving the next lemma we recall $h_n$ for convenience

$$h_n(\theta) = \begin{cases} \dfrac{v_n\theta}{\delta_n - \lambda_{n+1}} & \delta_n > \lambda_{n+1} \\[2ex] \dfrac{v_n\theta}{r_n} & \delta_n = \lambda_{n+1} \\[2ex] v_n\dfrac{\theta(r_n-1)!}{\lambda_{n+1}^{r_n}}\Phi(-\theta\alpha_{n+1}/\lambda_{n+1}, r_n, 1 - \delta_n/\lambda_{n+1}) & \delta_n < \lambda_{n+1}. \end{cases}$$

Then the following lemma will be of use.

**Lemma 4.** *With $f_n$ as in* Eq (13) *and $\kappa_n$ as in* Eq (16)*,*

$$\lim_{v_n \to 0} h_n\big(f_n(v_n)^{1/\gamma_n}\theta\big) = \kappa_n\theta^{\gamma_n}$$

*which implies that for $\delta_n > \lambda_{n+1}$*

$$\lim_{v_n \to 0} h_n(v_n^{-1}\theta) = \frac{\theta}{\delta_n - \lambda_{n+1}},$$

*for $\lambda_{n+1} = \delta_n$,*

$$\lim_{v_n \to 0} h_n(v_n^{-1}\theta) = \frac{\theta}{r_n},$$

*while for $\delta_n < \lambda_{n+1}$*

$$\lim_{v_n \to 0} h_n\big(v_n^{-1/\gamma_n}\log(v_n^{-1})^{-(r_n-1)/\gamma_n}\theta\big) = \frac{\phi_{n+1}^{1-\gamma_n}}{\lambda_{n+1}^{r_n}\gamma_n^{r_n-1}}\frac{\pi}{\sin\gamma_n\pi}\theta^{\gamma_n}.$$

*Proof.* Recall $\gamma_n = \delta_n/\delta_{n+1}$, $\phi_{n+1} = \lambda_{n+1}/\alpha_{n+1}$. The lemma is clearly true by the definition of $h_n(\theta)$ for $\delta_n > \lambda_{n+1}$ and $\delta_n = \lambda_{n+1}$.

We turn to the case of $\delta_n < \lambda_{n+1}$. For ease of notation we drop '$n$' subscripts and introduce $l_v = \log(v^{-1})$. From the definition of $h(\theta)$ in this case we see we require the limit of the Lerch transcendent for large first argument given in Lemma 3. Further, observe that for $a \in [0, 1]$, $\sin a\pi = \sin(1-a)\pi$. Hence, as $v \to 0$,

$$\Phi(-\theta v^{-1/\gamma}l_v^{-(r-1)/\gamma}\phi^{-1}, r, 1 - \gamma)$$

$$\sim \frac{\pi}{\sin\gamma\pi}\frac{1}{(\theta v^{-1/\gamma}l_v^{-(r-1)/\gamma}\phi^{-1})^{1-\gamma}}\frac{(\log[\theta v^{-1/\gamma}l_v^{-(r-1)/\gamma}\phi^{-1}])^{r-1}}{(r-1)!}$$

and so

$$
\begin{aligned}
h(v^{-1/\gamma}l_v^{-(r-1)/\gamma}\theta) \quad &\sim v^{1-1/\gamma}l_v^{-(r-1)/\gamma}\frac{\theta\Gamma(r)}{\lambda^r} \\
&\times \frac{\pi}{\sin\gamma\pi}\frac{1}{(\theta v^{-1/\gamma}l_v^{-(r-1)/\gamma}\phi^{-1})^{1-\gamma}}\frac{(\log[\theta v^{-1/\gamma}l_v^{-(r-1)/\gamma}\phi^{-1}])^{r-1}}{(r-1)!}.
\end{aligned}
$$

The $v$ factors outside of the logarithms immediately cancel, leaving the logarithmic factors. Collecting the logarithmic factors together, and recalling that $\Gamma(r_n) = (r_n - 1)!$, we have

$$
\begin{aligned}
h(v^{-1/\gamma}l_v^{-(r-1)/\gamma}\theta) \quad &\sim \frac{\phi^{1-\gamma}\theta^\gamma}{\lambda^r}\frac{\pi}{\sin\gamma\pi} \\
&\times l_v^{-(r-1)/\gamma}\frac{1}{(l_v^{-(r-1)/\gamma})^{1-\gamma}}[\log(\theta v^{-1/\gamma}l_v^{-(r-1)/\gamma})]^{r-1}.
\end{aligned}
$$

Notice that

$$
\begin{aligned}
[\log(\theta v^{-1/\gamma}l_v^{-(r-1)/\gamma})]^{r-1} \quad &= (\log(v^{-1/\gamma}) + \log(l_v^{-(r-1)/\gamma}\theta))^{r-1} \\
&\sim [\gamma^{-1}l_v]^{r-1}.
\end{aligned}
$$

Hence

$$
l_v^{-(r-1)/\gamma}\frac{1}{(l_v^{-(r-1)/\gamma})^{1-\gamma}}[\log(\theta v^{-1/\gamma}l_v^{-(r-1)/\gamma})]^{r-1} \to \gamma^{-(r-1)}.
$$

This leaves

$$
h(v^{-1/\gamma}l_v^{-(r-1)/\gamma}\theta) \to \frac{\phi^{1-\gamma}\theta^\gamma}{\lambda^r\gamma^{r-1}}\frac{\pi}{\sin\gamma\pi}
$$

as required.

We can now give the proof of Proposition 3:

*Proof of Proposition 3.* The base case is clear, we now argue by induction. We recall that

$$
\mathbb{E}[\exp(-V_{n+1}^*\theta)] = \mathbb{E}[\exp(-V_n^*h_n(\theta))].
$$

Hence

$$
\begin{aligned}
\mathbb{E}[\exp(-V_{n+1}^*\theta\mathcal{F}_n(\boldsymbol{v}))] \quad &= \mathbb{E}[\exp(-V_n^*h_n(\theta\mathcal{F}_n(\boldsymbol{v})))] \\
&= \mathbb{E}\left[\exp\left(-V_n^*h_n(\theta f_n(v_n)^{1/\gamma_n}\mathcal{F}_{n-1}(\boldsymbol{v})^{1/\gamma_n})\right)\right],
\end{aligned}
$$

where the relation between $\mathcal{F}_{n-1}(\boldsymbol{v})$ and $\mathcal{F}_n(\boldsymbol{v})$ given in Eq (15) was used. Thus

$$
\lim_{v_1\to 0}\ldots\lim_{v_n\to 0}\mathbb{E}[\exp(-V_{n+1}^*\theta\mathcal{F}_n(\boldsymbol{v}))]
$$

$$
= \lim_{v_1\to 0}\ldots\lim_{v_n\to 0}\mathbb{E}\left[\exp\left(-V_n^*h_n(\theta f_n(v_n)^{1/\gamma_n}\mathcal{F}_{n-1}(\boldsymbol{v})^{1/\gamma_n})\right)\right].
$$

Using Lemma 4, we have

$$\lim_{v_1 \to 0} \ldots \lim_{v_n \to 0} \mathbb{E}\left[\exp\left(- V_n^* h_n(\theta f_n(v_n)^{1/\gamma_n} \mathcal{F}_{n-1}(\boldsymbol{v})^{1/\gamma_n})\right)\right]$$

$$= \lim_{v_1 \to 0} \ldots \lim_{v_{n-1} \to 0} \mathbb{E}\left[\exp\left(- V_n^* \kappa_n [\theta \mathcal{F}_{n-1}(\boldsymbol{v})^{1/\gamma_n}]^{\gamma_n}\right)\right]$$

$$= \lim_{v_1 \to 0} \ldots \lim_{v_{n-1} \to 0} \mathbb{E}[\exp(-V_n^* \kappa_n \mathcal{F}_{n-1}(\boldsymbol{v})\theta^{\gamma_n})].$$

Using the induction hypothesis

$$\lim_{v_1 \to 0} \ldots \lim_{v_{n-1} \to 0} \mathbb{E}[\exp(-V_n^* \kappa_n \mathcal{F}_{n-1}(\boldsymbol{v})\theta^{\gamma_n})] = \mathbb{E}\left[\exp\left(- V_1^* (\kappa_n \theta^{\gamma_n})^{\delta_1/\delta_n} \prod_{i=1}^{n-1} \kappa_i^{\delta_1/\delta_i}\right)\right]$$

$$= \mathbb{E}\left[\exp\left(- V_1^* \theta^{\delta_1/\delta_{n+1}} \prod_{i=1}^{n} \kappa_i^{\delta_1/\delta_i}\right)\right].$$

We remark that when $\lambda_{i+1} \leq \delta_i$ (a fitness increase does not occur), we are not required to take the limit above on $v_i$—that is the statement of Proposition 3 is true without applying these limits.

Summarising thus far, we see

$$\lim_{v_1 \to 0} \ldots \lim_{v_n \to 0} \lim_{t \to \infty} \mathcal{F}_n(\boldsymbol{v})e^{-\delta_{n+1}t}t^{-(r_{n+1}-1)}Z_{n+1}^*(t)$$

has a Mittag-Leffler distribution with tail parameter $\delta_1/\delta_{n+1}$ and scale parameter

$$\left((\alpha_1/\lambda_1) \prod_{i=1}^{n} \kappa_i^{\delta_1/\delta_i}\right)^{\delta_{n+1}/\delta_1} = (\alpha_1/\lambda_1)^{\delta_{n+1}/\delta_1} \prod_{i=1}^{n} \kappa_i^{\delta_{n+1}/\delta_i}.$$

Separating into a time-dependent component this implies that

$$Z_{n+1}^*(t) \approx V_{n+1}^* e^{\delta_{n+1}t}t^{r_{n+1}-1} \tag{17}$$

with $V_{n+1}^*$ being Mittag-Leffler with tail parameter $\delta_1/\delta_{n+1}$ and scale parameter

$$\omega_{n+1} = (\alpha_1/\lambda_1)^{\delta_{n+1}/\delta_1} \mathcal{F}_n(\boldsymbol{v})^{-1} \prod_{i=1}^{n} \kappa_i^{\delta_{n+1}/\delta_i}. \tag{18}$$

If we consider the family of random variables $V_{n+1}^*$ then the scale parameters $\omega_{n+1}$ satisfy the following recursion

**Lemma 5.** *Set* $\omega_1 = \alpha_1/\lambda_1$, *then for* $n \geq 1$,

$$\omega_{n+1} = \begin{cases} \dfrac{v_n}{\delta_n - \lambda_{n+1}} \omega_n & \delta_n > \lambda_{n+1} \\ \dfrac{v_n}{r_n} \omega_n & \delta_n = \lambda_{n+1} \\ (v_n \log(v_n^{-1})^{r_n-1} \kappa_n \omega_n)^{\lambda_{n+1}/\delta_n} & \delta_n < \lambda_{n+1}, \end{cases} \tag{19}$$

*where* $\kappa_n$ *is defined in* Eq (16).

*Proof.* By Eq (18),

$$\omega_n = (\alpha_1/\lambda_1)^{\delta_n/\delta_1} \mathcal{F}_{n-1}(v)^{-1} \prod_{i=1}^{n-1} \kappa_i^{\delta_n/\delta_i}. \tag{20}$$

We now demonstrate that multiplying $\omega_n$ as given above, by the factors stated in Lemma 5 results in $\omega_{n+1}$ as expressed in Eq (18).

For the case of $\delta_n \geq \lambda_{n+1}$, $\kappa_n$ is either $(\delta_n - \lambda_{n+1})^{-1}$ for $\delta_n > \lambda_{n+1}$ or $r_n^{-1}$ for $\delta_n = \lambda_{n+1}$ (see the definition of $\kappa_n$ in Eq (16)). Hence, comprising both the cases of $\delta_n > \lambda_{n+1}$ and $\delta_n = \lambda_{n+1}$, we desire to show $v_n \kappa_n \omega_n = \omega_{n+1}$. Using Eq (20)

$$v_n \kappa_n \omega_n = v_n \kappa_n (\alpha_1/\lambda_1)^{\delta_n/\delta_1} \mathcal{F}_{n-1}(\boldsymbol{v})^{-1} \prod_{i=1}^{n-1} \kappa_i^{\delta_n/\delta_i}. \tag{21}$$

For $\delta_n \geq \lambda_{n+1}$, $\delta_n = \delta_{n+1}$. Moreover, $f_n(v_n) = v_n^{-1}$ (Eq (13)) and from Eq 15

$$\mathcal{F}_n(\boldsymbol{v})^{-1} = (f_n(v_n)\mathcal{F}_{n-1}(\boldsymbol{v}))^{-1} = v_n \mathcal{F}_{n-1}(\boldsymbol{v})^{-1}.$$

Thus, taking Eq (21), replacing each $\delta_n$ with $\delta_{n+1}$, and using the representation of $\mathcal{F}_n(\boldsymbol{v})^{-1}$,

$$v_n \kappa_n \omega_n \quad = \kappa_n (\alpha_1/\lambda_1)^{\delta_{n+1}/\delta_1} \mathcal{F}_n(\boldsymbol{v})^{-1} \prod_{i=1}^{n-1} \kappa_i^{\delta_{n+1}/\delta_i}.$$

Recognising that $\kappa_n = \kappa_n^{\delta_{n+1}/\delta_n}$ leads us to the desired form of $\omega_{n+1}$ as in Eq (18).

In the case of $\delta_n < \lambda_{n+1} = \delta_{n+1}$, we aim to demonstrate that $(v_n \log(v_n^{-1})^{r_n-1} \kappa_n \omega_n)^{\lambda_{n+1}/\delta_n}$ matches the expression for $\omega_{n+1}$ given in Eq (18). Again, using Eq (20),

$$(v_n \log(v_n^{-1})^{r_n-1} \kappa_n \omega_n)^{\lambda_{n+1}/\delta_n}$$

$$= \left[ v_n \log(v_n^{-1})^{r_n-1} \kappa_n (\alpha_1/\lambda_1)^{\delta_n/\delta_1} \mathcal{F}_{n-1}(\boldsymbol{v})^{-1} \prod_{i=1}^{n-1} \kappa_i^{\delta_n/\delta_i} \right]^{\lambda_{n+1}/\delta_n}$$

$$= \left[ (v_n \log(v_n^{-1})^{r_n-1})^{\delta_{n+1}/\delta_n} (\alpha_1/\lambda_1)^{\delta_{n+1}/\delta_1} \mathcal{F}_{n-1}(\boldsymbol{v})^{-\delta_{n+1}/\delta_n} \prod_{i=1}^{n} \kappa_i^{\delta_{n+1}/\delta_i} \right]. \tag{22}$$

For $\delta_n < \lambda_{n+1}$, $f_n(v_n) = v_n^{-1} \log(v_n^{-1})^{-(r_n-1)}$ (Eq (13)) and from Eq 15,

$$\mathcal{F}_n(\boldsymbol{v})^{-1} = (f_n(v_n)\mathcal{F}_{n-1}(\boldsymbol{v}))^{-\delta_{n+1}/\delta_n} = (v_n \log(v_n^{-1})^{r_n-1})^{\delta_{n+1}/\delta_n} \mathcal{F}_{n-1}(\boldsymbol{v})^{-\delta_{n+1}/\delta_n},$$

which combined with Eq (22) brings us to the desired form of $\omega_{n+1}$ as in Eq (18).

We summarise this approximate form of $Z_{n+1}^*(t)$ as a theorem, to emphasise that it is the culmination of the results in this section.

**Theorem 1** *For t large, and all $v_i$ small*

$$Z_{n+1}^*(t) \approx V_{n+1}^* e^{\delta_{n+1}t} t^{r_{n+1}-1}$$

*where $V_{n+1}^*$ is Mittag-Leffler distributed with tail parameter $\delta_1/\delta_{n+1}$ and scale parameter $\omega_{n+1}$ which satisfies the recurrence of Lemma 5.*

## Arrival times

We now turn to the time at which the type $n$ population arrives. Our limit results concerning this question are identical for both the original and approximate model, with only the parameters in the limit expressions changing. To avoid repeating results we introduce the superscript ∘, such that statements with variables with ∘ superscript are true for both models. Here, the

first time a cell arrives of type $n + 1$ is

$$\tau_{n+1}^\circ = \min\{t \geq 0 : Z_{n+1}^\circ(t) > 0\}.$$

It turns out $\tau_{n+1}^\circ$ can be appropriately centered using the following variables

$$\sigma_n = \delta_n^{-1} \log(v_n^{-1}), \quad m_n = \delta_n^{-1} \log(v_n^{-1}\sigma_n^{1-r_n}) \tag{23}$$

such that its distribution simplifies for small final seeding rates.

**Proposition 4.** As $v_n \to 0$,

$$\mathbb{P}(\tau_{n+1}^\circ - m_n > t) \to \mathbb{E}[\exp(-V_n^\circ e^{\delta_n t}/\delta_n)].$$

*Proof of Proposition 4.* We introduce $\rho_n = \delta_n^{-1} \log(\sigma_n^{r_n-1})$ so that $m_n = \sigma_n - \rho_n$. First let's condition on $\mathcal{Z}_n = (Z_n^\circ(s))_{s\in\mathbb{R}}$

$$\begin{aligned}
\mathbb{P}(\tau_{n+1}^\circ - (\sigma_n - \rho_n) > t | \mathcal{Z}_n) &= \exp\left(-v_n \int_0^{t+\sigma_n-\rho_n} Z_n^\circ(s)\, ds\right) \\
&= \exp\left(-v_n \int_{-(\sigma_n-\rho_n)}^t Z_n^\circ(u+\sigma_n-\rho_n)\, du\right)
\end{aligned}$$

Observe that $v_n Z_n^\circ(u + \sigma_n - \rho_n)$ can be expressed as

$$\frac{Z_n^\circ(u+\sigma_n-\rho_n)}{\exp(\delta_n(u+\sigma_n-\rho_n))(u+\sigma_n-\rho_n)^{r_n-1}}$$
$$\times\, v_n \exp(\delta_n(u+\sigma_n-\rho_n))(u+\sigma_n-\rho_n)^{r_n-1}.$$

As $v_n \to 0$ the first factor above converges to $V_n^\circ$. The second factor may be expressed as

$$e^{\delta_n u} \frac{(u+\sigma_n-\rho_n)^{r_n-1}}{\sigma_n^{r_n-1}}$$

which converges to $e^{\delta_n u}$ as $v_n \to 0$. Hence $v_n Z_n^\circ(u+\sigma_n-\rho_n) \to V_n^\circ e^{\delta_n u}$.

Propositions 1 and 2 imply that for any realisation we may find small enough $x$ such that for $v_n \leq x$

$$Z_n^\circ(u+\sigma_n-\rho_n) \leq 2V_n^\circ e^{\delta_n u} u^{r_n-1}$$

which is integrable over $(-\infty, t]$. Using dominated convergence we have the claimed result.

We know that with $\delta_n = \lambda_1$, $V_n^\circ$ has an exponential distribution, and so the limit distribution for $\tau_{n+1}^\circ$ may be immediately obtained [11]. If there are fitness increases, we turn to our small mutation results for the approximate model.

For the remainder of this section we discuss only results for the approximate model. The below results also hold for the original branching processes if the running-max fitness does not increase, i.e. $\delta_n = \lambda_1$.

Thus with $\mathcal{F}_{n-1}(v)$ as in Eq 14, and using Proposition 3, we see that:

**Corollary 3**.

$$\lim_{v_1 \to 0} \ldots \lim_{v_n \to 0} \mathbb{P}(\tau_{n+1}^* - m_n - \delta_n^{-1} \log \mathcal{F}_{n-1}(\boldsymbol{v}) > t)$$

$$= \mathbb{E}\left[\exp\left(- V_1^* e^{\delta_1 t} \delta_n^{-\delta_1/\delta_n} \prod_{i=1}^{n-1} \kappa_i^{\delta_1/\delta_i}\right)\right]$$

$$= \left(1 + [(\lambda_1/\alpha_1)\delta_n^{\delta_1/\delta_n}]^{-1} e^{\delta_1 t} \prod_{i=1}^{n-1} \kappa_i^{\delta_1/\delta_i}\right)^{-1}$$

*Proof.* From Proposition 4

$$\lim_{v_1 \to 0} \ldots \lim_{v_n \to 0} \mathbb{P}(\tau_{n+1}^* - m_n - \delta_n^{-1} \log \mathcal{F}_{n-1}(\boldsymbol{v}) > t)$$

$$= \lim_{v_1 \to 0} \ldots \lim_{v_{n-1} \to 0} \mathbb{E}[\exp(-V_n^* \mathcal{F}_{n-1}(\boldsymbol{v}) e^{\delta_n t} / \delta_n)].$$

While from Proposition 3,

$$\lim_{v_1 \to 0} \ldots \lim_{v_{n-1} \to 0} \mathbb{E}[\exp(-V_n^* \mathcal{F}_{n-1}(\boldsymbol{v}) e^{\delta_n t} / \delta_n))]$$

$$= \mathbb{E}\left[\exp\left(- V_1^* (e^{\delta_n t}/\delta_n)^{\delta_1/\delta_n} \prod_{i=1}^{n-1} \kappa_i^{\delta_1/\delta_i}\right)\right]$$

$$= \left(1 + [(\lambda_1/\alpha_1)\delta_n^{\delta_1/\delta_n}]^{-1} e^{\delta_1 t} \prod_{i=1}^{n-1} \kappa_i^{\delta_1/\delta_i}\right)^{-1}$$

This implies that for small mutation rates

$$\mathbb{P}(\tau_{n+1}^* > t) \quad = \mathbb{P}(\tau_{n+1}^* - m_n - \delta_n^{-1} \log \mathcal{F}_{n-1}(\boldsymbol{v}) > t - m_n - \delta_n^{-1} \log \mathcal{F}_{n-1}(\boldsymbol{v}))$$

$$\approx \mathbb{E}\left[\exp\left(- V_1^* \delta_n^{-\delta_1/\delta_n} e^{\delta_1 t} \mathcal{F}_{n-1}(\boldsymbol{v})^{-\delta_1/\delta_n} e^{-\delta_1 m_n} \prod_{i=1}^{n-1} \kappa_i^{\delta_1/\delta_i}\right)\right]$$

$$= \left(1 + \delta_n^{-\delta_1/\delta_n} e^{\delta_1 t} (\alpha_1/\lambda_1) \mathcal{F}_{n-1}(\boldsymbol{v})^{-\delta_1/\delta_n} e^{-\delta_1 m_n} \prod_{i=1}^{n-1} \kappa_i^{\delta_1/\delta_i}\right)^{-1}$$

Recall that

$$\omega_n = (\alpha_1/\lambda_1)^{\delta_n/\delta_1} \mathcal{F}_{n-1}(\boldsymbol{v})^{-1} \prod_{i=1}^{n-1} \kappa_i^{\delta_n/\delta_i},$$

and that by the definition of $m_n$,

$$e^{-\delta_1 m_n} = \exp\left[-\frac{\delta_1}{\delta_n} \log[v_n^{-1} (\delta_n^{-1} \log(v_n^{-1}))^{-(r_n-1)}]\right] = v_n^{\delta_1/\delta_n} (\delta_n^{-1} \log(v_n^{-1}))^{(r_n-1)\delta_1/\delta_n}.$$

Hence

$$\mathbb{P}(\tau_{n+1}^* > t) \approx \left[1 + e^{\delta_1 t} \left(\frac{\omega_n v_n (\delta_n^{-1} \log(v_n^{-1}))^{(r_n-1)}}{\delta_n}\right)^{\delta_1/\delta_n}\right]^{-1}.$$

Defining

$$t_{1/2}^{(n+1)} = \frac{1}{\delta_n} \log \frac{\delta_n}{\omega_n v_n [\delta_n^{-1} \log(v_n^{-1})]^{r_n-1}}$$

we see that $\tau_{n+1}^*$ has a logistic distribution with scale parameter $\delta_1^{-1}$ and median $t_{1/2}^{(n+1)}$

$$\mathbb{P}(\tau_{n+1}^* > t) \approx \left[1 + e^{\delta_1(t - t_{1/2}^{(n+1)})}\right]^{-1} \tag{24}$$

The median times satisfy the following recurrence:

**Lemma 6.** *Set*

$$t_{1/2}^{(2)} = \frac{1}{\delta_1} \log \frac{\delta_1^2}{\alpha_1 v_1}.$$

*Then for $n \geq 2$*

$$t_{1/2}^{(n+1)} = t_{1/2}^{(n)} + \begin{cases} \frac{1}{\delta_n} \log \frac{(\delta_n - \lambda_n)}{v_n} \left[\frac{\log(v_{n-1}^{-1})}{\log(v_n^{-1})}\right]^{r_n-1} & \delta_{n-1} > \lambda_n \\[2ex] \frac{1}{\delta_n} \log \frac{r_{n-1} \delta_{n-1}}{v_n} \frac{[\log(v_{n-1}^{-1})]^{r_{n-1}-1}}{[\log(v_n^{-1})]^{r_n-1}} & \delta_{n-1} = \lambda_n \\[2ex] \frac{1}{\delta_n} \log \frac{\delta_n}{v_n [\delta_n^{-1} \log(v_n^{-1})]^{r_n-1}} - \frac{1}{\delta_{n-1}} \log(\delta_{n-1}^{r_{n-1}} \kappa_{n-1}) & \delta_{n-1} < \lambda_n \end{cases} \tag{25}$$

*Proof.* We start with $\lambda_n < \delta_{n-1}$, in which case $\omega_n = \frac{v_{n-1}}{\delta_{n-1} - \lambda_n} \omega_{n-1}$, and $\delta_{n-1} = \delta_n$, $r_n = r_{n-1}$, thus

$$\begin{aligned} t_{1/2}^{(n+1)} &= \frac{1}{\delta_n} \log \frac{\delta_n(\delta_{n-1} - \lambda_n)}{v_n [\delta_n^{-1} \log(v_n^{-1})]^{r_n-1} v_{n-1} \omega_{n-1}} \\[2ex] &= \frac{1}{\delta_n} \log \frac{(\delta_{n-1} - \lambda_n)}{v_n [\delta_n^{-1} \log(v_n^{-1})]^{r_n-1}} + \frac{1}{\delta_n} \log \frac{\delta_n}{v_{n-1} \omega_{n-1}} \\[2ex] &= \frac{1}{\delta_n} \log \frac{(\delta_{n-1} - \lambda_n)}{v_n} \frac{[\delta_{n-1}^{-1} \log(v_{n-1}^{-1})]^{r_{n-1}-1}}{[\delta_n^{-1} \log(v_n^{-1})]^{r_n-1}} \\[2ex] &\quad + \frac{1}{\delta_n} \log \frac{\delta_n}{v_{n-1} \omega_{n-1} [\delta_{n-1}^{-1} \log(v_{n-1}^{-1})]^{r_{n-1}-1}} \\[2ex] &= \frac{1}{\delta_n} \log \frac{(\delta_{n-1} - \lambda_n)}{v_n} \left[\frac{\log(v_{n-1}^{-1})}{\log(v_n^{-1})}\right]^{r_n-1} \\[2ex] &\quad + \frac{1}{\delta_{n-1}} \log \frac{\delta_{n-1}}{v_{n-1} \omega_{n-1} [\delta_{n-1}^{-1} \log(v_{n-1}^{-1})]^{r_{n-1}-1}} \\[2ex] &= \frac{1}{\delta_n} \log \frac{(\delta_{n-1} - \lambda_n)}{v_n} \left[\frac{\log(v_{n-1}^{-1})}{\log(v_n^{-1})}\right]^{r_n-1} + t_{1/2}^{(n)} \\[2ex] &= \frac{1}{\delta_n} \log \frac{(\delta_n - \lambda_n)}{v_n} \left[\frac{\log(v_{n-1}^{-1})}{\log(v_n^{-1})}\right]^{r_n-1} + t_{1/2}^{(n)}. \end{aligned}$$

For the case of $\lambda_n = \delta_{n-1}$, then $\omega_n = v_{n-1}\omega_{n-1}/r_{n-1}$ and $\delta_n = \delta_{n-1}$, $r_n = r_{n-1} + 1$, thus

$$
\begin{aligned}
t_{1/2}^{(n+1)} \quad &= \frac{1}{\delta_n}\log\frac{\delta_n r_{n-1}}{v_n[\delta_n^{-1}\log(v_n^{-1})]^{r_{n-1}}v_{n-1}\omega_{n-1}} \\
&= \frac{1}{\delta_n}\log\frac{r_{n-1}}{v_n}\frac{[\delta_{n-1}^{-1}\log(v_{n-1}^{-1})]^{r_{n-1}-1}}{[\delta_n^{-1}\log(v_n^{-1})]^{r_{n-1}}} \\
&\quad + \frac{1}{\delta_n}\log\frac{\delta_n}{v_{n-1}\omega_{n-1}[\delta_{n-1}^{-1}\log(v_{n-1}^{-1})]^{r_{n-1}-1}} \\
&= \frac{1}{\delta_n}\log\frac{r_{n-1}\delta_{n-1}}{v_n}\frac{[\log(v_{n-1}^{-1})]^{r_{n-1}-1}}{[\log(v_n^{-1})]^{r_{n-1}}} \\
&\quad + \frac{1}{\delta_{n-1}}\log\frac{\delta_{n-1}}{v_{n-1}\omega_{n-1}[\delta_{n-1}^{-1}\log(v_{n-1}^{-1})]^{r_{n-1}-1}} \\
&= \frac{1}{\delta_n}\log\frac{r_{n-1}\delta_{n-1}}{v_n}\frac{[\log(v_{n-1}^{-1})]^{r_{n-1}-1}}{[\log(v_n^{-1})]^{r_{n-1}}} + t_{1/2}^{(n)}.
\end{aligned}
$$

Turning to the case of $\lambda_n > \delta_{n-1}$, we have $\omega_n = (\omega_{n-1}v_{n-1}\log(v_{n-1}^{-1})^{r_{n-1}-1}\kappa_{n-1})^{\lambda_n/\delta_{n-1}}$, or alternatively

$$
\omega_n\delta_{n-1}^{-(r_{n-1}-1)\lambda_n/\delta_{n-1}} = [\omega_{n-1}v_{n-1}[\delta_{n-1}^{-1}\log(v_{n-1}^{-1})]^{r_{n-1}-1}\kappa_{n-1}]^{\lambda_n/\delta_{n-1}}.
$$

and we also have $\delta_n = \lambda_n$ and $r_n = r_{n-1}$. Similarly to before

$$
\begin{aligned}
t_{1/2}^{(n+1)} \quad &= \frac{1}{\delta_n}\log\frac{\delta_n}{v_n[\delta_n^{-1}\log(v_n^{-1})]^{r_n-1}} + \frac{1}{\delta_n}\log\frac{\delta_{n-1}^{-(r_{n-1}-1)\lambda_n/\delta_{n-1}}}{\omega_n\delta_{n-1}^{-(r_{n-1}-1)\lambda_n/\delta_{n-1}}} \\
&= \frac{1}{\delta_n}\log\frac{\delta_n}{v_n[\delta_n^{-1}\log(v_n^{-1})]^{r_n-1}} + \frac{1}{\delta_n}\log\delta_{n-1}^{-(r_{n-1}-1)\delta_n/\delta_{n-1}} \\
&\quad + \frac{1}{\delta_n}\log\frac{\delta_{n-1}^{\delta_n/\delta_{n-1}}}{[\omega_{n-1}v_{n-1}[\delta_{n-1}^{-1}\log(v_{n-1}^{-1})]^{r_{n-1}-1}]^{\delta_n/\delta_{n-1}}} \\
&\quad + \frac{1}{\delta_n}\log\frac{1}{(\delta_{n-1}\kappa_{n-1})^{\delta_n/\delta_{n-1}}} \\
&= \frac{1}{\delta_n}\log\frac{\delta_n}{v_n[\delta_n^{-1}\log(v_n^{-1})]^{r_n-1}} + \frac{1}{\delta_{n-1}}\log\frac{1}{\delta_{n-1}^{r_{n-1}}\kappa_{n-1}} + t_{1/2}^{(n)}
\end{aligned}
$$

We summarise this approximate distribution of $\tau_{n+1}^*$ as a theorem, to emphasise that it is the culmination of the results in this section.

**Theorem 2**. *For $t \geq 0$ and all $v_i$ small*

$$
\mathbb{P}(\tau_{n+1}^* > t) \approx \left[1 + e^{\delta_1(t - t_{1/2}^{(n+1)})}\right]^{-1}.
$$

*where the median times $t_{1/2}^{(n+1)}$ which satisfies the recurrence of Lemma 6.*

**Remark 1** In the above results we take the ordered limit $\lim_{v_1 \to 0} \ldots \lim_{v_n \to 0}$ for two technical reasons:

(i) In the proof of Proposition 4 we used the almost sure convergence of the scaled type $n$ cell number, that is Proposition 2. As the type $n$ populations' growth is unaffected by the value of $v_n$, no issues arise. However, the type $n$'s growth is affected by $v_1, \ldots, v_{n-1}$, and so almost sure convergence of cell numbers would not hold when simultaneously sending these mutation rates to 0, thus invalidating our proof strategy.

(ii) We build our understanding of the limit random variable $V_{n+1}^*$ from the distribution of $V_n^*$, as seen in Corollary 2. Small mutation rate limits were required to circumvent the complexity introduced by the Lerch transcendent in $h_n(\theta)$, and then ultimately in the composite function—composing all $h_i$—in Eq (12). In the composite function of Eq (12), the function $h_{i+1}$ is applied before $h_i$, hence the mutation rate ordering.

This specific ordering may have consequences on higher order details; for example in Eq (24), the final mutation rate $v_n$ is privileged, appearing in the $\log(v_n^{-1})$ term. In other limits, e.g. all mutation rates are equal, this term may alter. On the other hand, when considering $\tau_{n+1}$, we wait for the first mutation of type $n + 1$, whereas multiple mutations may occur from type $i \to i + 1$ for $i = \ldots, n - 1$; so the $\log(v_n^{-1})$ might remain in alternative limit orders. However, for practical scenarios we do not expect this feature to considerably impact results; this may be seen by the considering the median time $t_{1/2}^{(n)}$, where it's clear that the privileged term acts as a higher order loglog correction to the leading behaviour.

## Supporting information

**S1 Fig. Comparison of limiting logistic distribution for hitting times with stochastic simulations.** Empirical cumulative distribution of the arrival times of types 1–3 obtained from simulations of the exact model versus the cumulative distribution function corresponding to the logistic distribution of Eq 4. Birth/death parameters: A (net growth rate decreases then increases), $\alpha_1 = \alpha_2 = 1, \alpha_3 = 1.4, \beta_1 = \beta_3 = 0.3, \beta_2 = 1.5$; B, D (net growth rate increases then decreases); $\alpha_1 = \alpha_3 = 1, \alpha_2 = 1.4, \beta_1 = \beta_2 = 0.3, \beta_3 = 1.5$; C (neutral), $\alpha_1 = \alpha_2 = \alpha_3 = 1, \beta_1 = \beta_2 = \beta_3 = 0.3$. Mutation rates: A, B, C, $v_1 = v_2 = v_3 = 0.01$; D, $v_1 = v_2 = v_3 = 0.001$. Number of simulations: A, B, C; 1000 simulations; D, 100 simulations.
(TIF)

**S1 Text. Statistical methods for $n$-mutation fluctuation assay.**
(PDF)

## Acknowledgments

We are grateful to Adri B. Olde Daalhuis for his help with Lemma 3, and to Martin Reijns for discussions on fluctuation assay experiments.

## Author Contributions

**Conceptualization:** Michael D. Nicholson, David Cheek, Tibor Antal.

**Formal analysis:** Michael D. Nicholson, David Cheek, Tibor Antal.

**Investigation:** Michael D. Nicholson, David Cheek, Tibor Antal.

**Methodology:** Michael D. Nicholson, David Cheek, Tibor Antal.

**Software:** Michael D. Nicholson, Tibor Antal.

**Visualization:** Michael D. Nicholson, David Cheek, Tibor Antal.

**Writing – original draft:** Michael D. Nicholson, David Cheek, Tibor Antal.

**Writing – review & editing:** Michael D. Nicholson, David Cheek, Tibor Antal.

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
