## [Decision Letter · Decision Letter 0]

2 Dec 2022

Dear Dr Nicholson,

Thank you very much for submitting your manuscript "Mutation accumulation in exponentially growingpopulations" for consideration at PLOS Computational Biology.

As with all papers reviewed by the journal, your manuscript was reviewed by members of the editorial board and by several independent reviewers. In light of the reviews (below this email), we would like to invite the resubmission of a significantly-revised version that takes into account the reviewers' comments.

This is a nice manuscript and set of results. The reviewers have raised some concerns about the strengthening the connection of this work to the biological applications, in terms of both the model assumptions and the applicability of results. Please address these concerns and the other items raised in the reports.

We cannot make any decision about publication until we have seen the revised manuscript and your response to the reviewers' comments. Your revised manuscript is also likely to be sent to reviewers for further evaluation.

Sincerely,

Jasmine Foo

Guest Editor

PLOS Computational Biology

Natalia Komarova

Section Editor

PLOS Computational Biology

This is a nice manuscript and set of results. The reviewers have raised some concerns about the strengthening the connection of this work to the biological applications, in terms of both the model assumptions and the applicability of results. Please address these concerns and the other items raised in the reports.

Reviewer's Responses to Questions

**Comments to the Authors:**

Reviewer #1: Review uploaded as an attachment

Reviewer #2: Review is uploaded as an attachment.

**Have the authors made all data and (if applicable) computational code underlying the findings in their manuscript fully available?**

Reviewer #1: Yes

Reviewer #2: None

PLOS authors have the option to publish the peer review history of their article (what does this mean?). If published, this will include your full peer review and any attached files.

Reviewer #1: No

Reviewer #2: No
---

## [Decision Letter · Decision Letter 1]

8 Jun 2023

Dear Dr Nicholson,

Thank you very much for submitting your manuscript "Sequential mutations in exponentially growing populations" for consideration at PLOS Computational Biology. As with all papers reviewed by the journal, your manuscript was reviewed by members of the editorial board and by several independent reviewers. The reviewers appreciated the attention to an important topic. Based on the reviews, we are likely to accept this manuscript for publication, providing that you modify the manuscript according to the review recommendations.

Thank you for your efforts in revising this manuscript. Please address the minor revisions and typos pointed out by Reviewer 2.

Sincerely,

Jasmine Foo

Guest Editor

PLOS Computational Biology

Natalia Komarova

Section Editor

PLOS Computational Biology

Thank you for your efforts in revising this manuscript. Please address the minor revisions and typos pointed out by Reviewer 2.

Reviewer's Responses to Questions

**Comments to the Authors:**

Reviewer #1: The authors have addressed all my previous concerns and I am happy with the current version to be published in PLoS CB. .

Reviewer #2: Review is uploaded as an attachment.

**Have the authors made all data and (if applicable) computational code underlying the findings in their manuscript fully available?**

Reviewer #1: Yes

Reviewer #2: Yes

PLOS authors have the option to publish the peer review history of their article (what does this mean?). If published, this will include your full peer review and any attached files.

Reviewer #1: No

Reviewer #2: No

Figure Files:

Data Requirements:

Reproducibility:

References:

---

## [Editor Report · Decision Letter 2]

21 Jun 2023

Dear Dr Nicholson,

We are pleased to inform you that your manuscript 'Sequential mutations in exponentially growing populations' has been provisionally accepted for publication in PLOS Computational Biology.

Best regards,

Jasmine Foo

Guest Editor

PLOS Computational Biology

Natalia Komarova

Section Editor

PLOS Computational Biology

---

## [Editor Report · Acceptance letter]

5 Jul 2023

PCOMPBIOL-D-22-01344R2 

Sequential mutations in exponentially growing populations

Dear Dr Nicholson,

I am pleased to inform you that your manuscript has been formally accepted for publication in PLOS Computational Biology. Your manuscript is now with our production department and you will be notified of the publication date in due course.

With kind regards,

Zsofi Zombor
